# Physical principles of retroviral integration in the human genome

D. Michieletto[1], M. Lusic[2], D. Marenduzzo[1] & E. Orlandini [ID] [3]

Certain retroviruses, including HIV, insert their DNA in a non-random fraction of the host genome via poorly understood selection mechanisms. Here, we develop a biophysical model for retroviral integration as stochastic and quasi-equilibrium topological reconnections between polymers. We discover that physical effects, such as DNA accessibility and elasticity, play important and universal roles in this process. Our simulations predict that integration is favoured within nucleosomal and flexible DNA, in line with experiments, and that these biases arise due to competing energy barriers associated with DNA deformations. By considering a long chromosomal region in human T-cells during interphase, we discover that at these larger scales integration sites are predominantly determined by chromatin accessibility. Finally, we propose and solve a reaction-diffusion problem that recapitulates the distribution of HIV hot-spots within T-cells. With few generic assumptions, our model can rationalise experimental observations and identifies previously unappreciated physical contributions to retroviral integration site selection.

[1] SUPA, School of Physics and Astronomy, University of Edinburgh, Peter Guthrie Tait Road, Edinburgh EH9 3FD, UK. [2] Department of Infectious Diseases, Integrative Virology, Heidelberg University Hospital and German Center for Infection Research, Im Neuenheimer Feld 344, 69120 Heidelberg, Germany. [3] Dipartimento di Fisica e Astronomia e Sezione INFN, Universitá di Padova, Via Marzolo 8, 35131 Padova, Italy. Correspondence and requests for materials should be addressed to D.M. (email: davide.michieletto@ed.ac.uk) or to D.M. (email: dmarendu@ph.ed.ac.uk)

Retroviruses are pathogens that infect organisms by inserting their DNA within the genome of the host. Once integrated, they exploit the transcription machinery already in place to proliferate and propagate themselves into other cells or organisms[1–3]. This strategy ingrains the viral DNA in the host cell and it ensures its transmission to the daughter cells; about 5–10% of the human genome is made up by ancient retroviral DNA, mutated in such a way that it is no longer able to replicate itself[4,5]. Whilst many retroviruses clearly pose a danger to health, they are also potentially appealing for clinical medicine, as they can be used as vectors for gene therapies[1,6,7].

Experiments have provided a wealth of important observations on the mechanisms through which retroviruses work. Classical experiments have shown that the retroviral integration complex (or 'intasome') displays a marked tendency to target bent DNA regions and in particular those wrapped around histones rather than naked DNA[8–14]. This is clearly advantageous for retroviruses which target eukaryotes, since their DNA is extensively packaged into chromatin[2]. More recent experiments also suggest that the integration sites displayed by most classes of retroviruses are correlated with the underlying chromatin state[15]. For instance, gammaretroviruses, deltaretroviruses and lentiviruses—including HIV—display a strong preference to insert their DNA into transcriptionally active chromatin[15–17]. Importantly, the preference for transcriptionally active regions remains significantly non-random even after knockout of known tethering factors such as LEDGF/p75[15,17–19], or double knockdown of LEDGF and other putative protein chaperones[17].

In stark contrast with the abundance of experiments aimed at studying the roles played by system-specific co-factors in retroviral integration (see ref. [15] for a review), there is a distinct lack of models to address generic principles of this complex problem. Such an approach may provide a useful complement to existing and future experiments and may shed light into the universal, i.e. non-system-specific, behaviour of retroviral integration. To fill this gap, here we propose a generic biophysical model for retroviral integration in host genomes, focussing on the case of HIV for which there is extensive literature and experimental evidence. We first introduce and study a framework in which retroviral DNA and host genomes are modelled as semi-flexible polymers, and integration events are accounted for by performing local stochastic recombination moves between 3D-proximal polymer segments. Then, at larger scales, we formulate and solve a reaction–diffusion problem to study HIV integration within the nuclear environment of human cells.

At all scales considered, ranging from that of single nucleosomes to that of the cell nucleus, our model compares remarkably well with experiments, both qualitatively and quantitatively. In light of this, we argue that simple physical features, such as DNA elasticity and large-scale chromosome folding, may cover important and complementary roles to those of known co-factors in dictating retroviral integration patterns.

## Results

**A quasi-equilibrium model for retroviral integration.** When retroviruses, and in particular HIV, enter the nucleus of a cell, they do so in the form of a pre-integration complex (PIC)[20]. This is made by the viral DNA (vDNA), the integrase (IN) enzyme, joining the long-terminal-repeats (LTRs) into the intasome structure, and a number of host enzymes that facilitate nuclear import and trafficking[21].

In an effort to simplify a model for such process, here we choose to only account for vDNA and IN, as these are the elements that are necessary and sufficient to perform successful integrations in vitro on a target DNA (tDNA)[9]. Both vDNA and

tDNA are treated as semi-flexible bead-spring chains, a broadly employed polymer model for DNA and chromatin[22–25]. These polymers are made of beads of size $\sigma$ and with persistence length $l_p$ typically set to 50 nm for DNA[26] and 30 nm for chromatin[27] (Fig. 1a). The dynamics of the chains are evolved by performing molecular dynamics (MD) simulations in Brownian mode, which implicitly accounts for the solvent; this means that vDNA and tDNA explore the space diffusively, and that the vDNA searches for its integration target via 3D diffusion.

The IN enzyme mediates integration through a complex pathway[12,28]. Yet, here we are interested in formulating a simple model that can capture the essential physics of the process. We thus condense vDNA insertion into one stochastic step: the swap of two polymer bonds which are transiently close in 3D space (specifically, within $R_c = 2\sigma = 5$ nm, see Fig. 1b). If successful, the vDNA is inserted into the tDNA and it is irreversibly trapped in place; if rejected, the vDNA is not inserted into the host and it resumes its diffusive search. [Accounting for the precise position of the intasome along the vDNA does not change our results and we discuss this refinement in Supplementary Note 3].

Because integration of vDNA into a tDNA can be performed in vitro in absence of ATP[7,8] we argue that the IN enzyme must work in thermal equilibrium. For this reason, we choose to assign an equilibrium acceptance probability to the integration move by computing the total internal energy $E$ of the polymer configurations before ($\Omega$) and after ($\Omega'$) the reconnection (Fig. 1b). This energy is made of contributions from the bending of the chains, stretching of the bonds and steric interactions. The energy difference $\Delta E = E(\Omega') - E(\Omega)$ is then used to assign the (Metropolis) probability $p = \min\left\{1, e^{-\Delta E/k_B T}\right\}$ for accepting or rejecting the integration attempt. Notice that because a successful integration event is irreversible, in reality this process is only in quasi-equilibrium as it violates detailed balance.

Whilst our stochastic quasi-equilibrium model does not reproduce the correct sequence of molecular events leading to integration[28], it correctly captures the integration kinetics at longer timescales. This is because such kinetics depend on steric interactions and the energy barrier associated with integration, both included in our model. As the host DNA needs to be severely bent upon integration[14,28], and as this deformation expends energy that is not provided by ATP[7], we conjecture that the IN enzyme will effectively probe the substrate for regions with lower energy barriers against local bending deformations. [We mention that in refs. [13,14] the authors considered a 1D physical model where the probability of integration is equal to the Boltzmann weight of the elastic energy of DNA, equilibrated after insertion. On the contrary, here we consider a dynamic quasi-equilibrium stochastic process in 3D where the energy barrier against local deformations and diffusive search are the main determinants of integration profiles].

**The nucleosome is a geometric catalyst for integration.** Extensive in vitro experiments on HIV integration in artificially designed DNA sequences revealed that the IN enzyme displays a pronounced preference for flexible or intrinsically curved DNA sequences[9,10] and that chromatinised substrates are more efficiently targeted than naked DNA[8]. The affinity to histone-bound DNA is counterintuitive as the nucleosomal structure may be thought to hinder intasome accessibility to the underlying DNA[12].

To rationalise these findings, we simulate the integration of a short vDNA (40 beads or 320 bp) within a DNA sequence made of 100 beads (or 800 bp) in which the central 20 beads (160 bp) are wrapped in a nucleosomal structure. [The precise lengths of vDNA and tDNA do not change our results as the integration

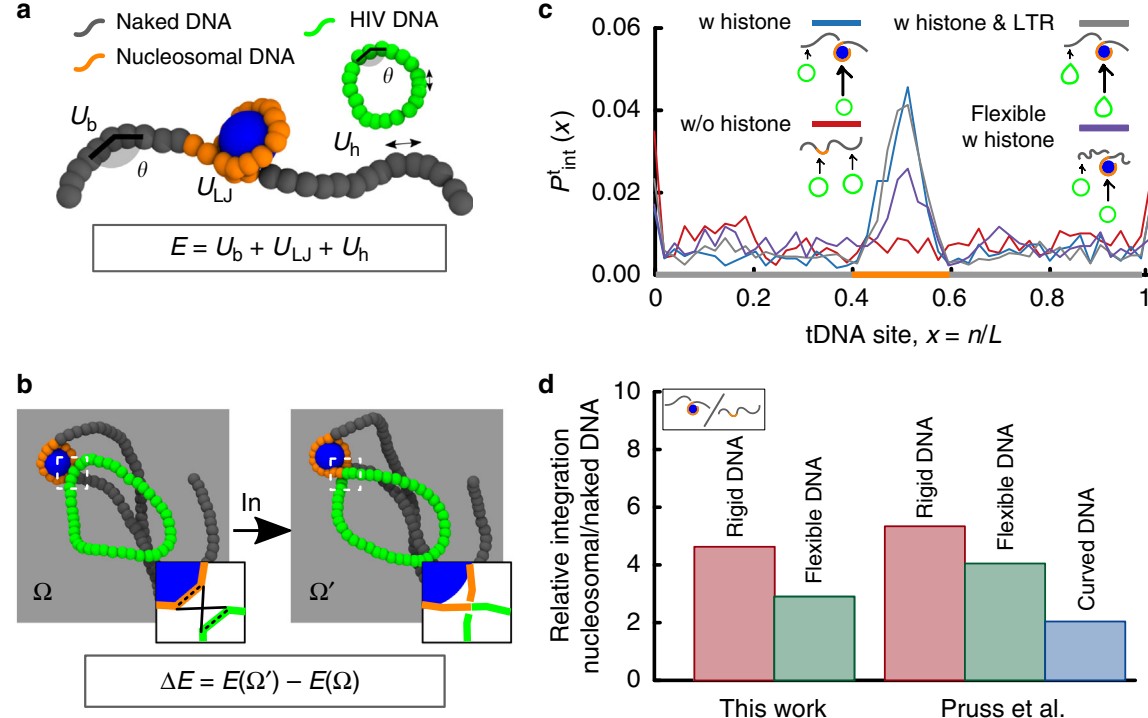

**Fig. 1** DNA elasticity biases HIV integration in nucleosomes. **a** Model for tDNA and vDNA as diffusing bead-spring polymers with bending rigidity. The potentials associated with bending ($U_b(\theta)$), steric/attractive interactions ($U_{LJ}$) and stretching of the bonds ($U_h$) contribute to the energy $E$ of a given configuration. **b** A quasi-equilibrium stochastic integration event takes into account the energy before and after the integration event ($E(\Omega)$ and $E(\Omega')$, respectively) to determine a successful integration probability $p = \min\{\exp(-\Delta E/k_BT), 1\}$. **c** The integration profile $P_{int}(x)$ as a function of the relative tDNA site, $x = n/L$, displays a ~4-fold enhancement in the region wrapped around the histone-like protein. The same behaviour is observed when a kinked site (corresponding to the intasome flanked by LTR) is included in the vDNA. Considering flexible tDNA ($l_p = 30$ nm) weakens this preference. **d** Direct quantitative comparison of relative integration enhancement with the data reported in ref. [8]. The integration profiles in **c** are averages over 1000 independent simulations and the dynamics of the simulated process can be seen in Supplementary Movies 1 and 2. Source data are provided as a Source Data file

moves are performed locally, see Supplementary Note 1 and Supplementary Fig. 1] The nucleosome is modelled by setting a short-ranged attraction between the central segment (orange in Fig. 1a) and a histone-like protein of size $\sigma_h = 3\sigma = 7.5$ nm[22] (see Supplementary Note 2). In our simulations, tDNA and the histone-like protein diffuse within a confined region of space and self-assemble in a nucleosome as seen in Fig. 1a. We then allow the diffusing vDNA to integrate anywhere along the substrate and compute the probability of observing an integration event, $P_{int}(x)$, as a function of the genomic position $x$. As shown in Fig. 1c, $P_{int}(x)$ displays a ~4-fold increase within the nucleosome and is instead random, i.e. uniform, in naked DNA (Fig. 1c). In all cases, $P_{int}(x)$ increases at the ends of the host polymer, as integration there entails a smaller bending energy barrier.

These results can be explained by the following argument: tDNA bound to histones is highly bent and thus the energy barrier associated with its deformation leading to integration is smaller than outside the nucleosome. We also point out that if all nucleosomal segments were fully wrapped, we would expect a flat-top integration probability rather than one peaked at the dyad. Because in our model nucleosomes are dynamic and may partially unravel, the most likely segment to be histone-bound at any time is, by symmetry, the central dyad thus explaining the shape of the integration profile (see Fig. 1c).

We conclude this section by directly comparing our findings with those from refs. [8,9] (Fig. 1d). First, we notice that the ~4-fold enhancement of nucleosomal integration predicted by our model is in remarkable agreement with the values reported in ref. [8] for rigid DNA substrates. Second, the same authors show that this

bias is weakened by considering flexible or intrinsically curved DNA substrates[8]. Motivated by this finding, we repeat our simulations using a more flexible substrate with $l_p = 30$ nm and observe the same weakening (see Fig. 1c, d). This behaviour fits within our simple argument: since more flexible (or curved) DNA segments display a much smaller conformational energy when wrapped around histones, the difference in bending energy within and outside nucleosomal regions is largely reduced.

It is finally important to stress that in refs. [8,9], the experiments were performed in absence of other enzymatic co-factors. Hence, the observed bias must be solely due to the viral IN enzyme. This is fully consistent with our results, which show that the nucleosome acts as a geometric catalyst for retroviral integration.

**Retroviral integration in supercoiled DNA.** DNA's torsional rigidity gives rise to non-trivial features, such as supercoiling, which may generically affect the retroviral insertion process. Our model is currently not equipped to correctly capture twist deformations, although these could be accounted for in future refinements[29]. At the same time, it is useful to discuss some experimental facts and theoretical expectations on the role of supercoiling in this process.

First, long-standing experiments report that HIV and Moloney murine leukaemia virus integration profiles display a 10 bp periodicity along nucleosomal DNA that coincide with the sites where DNA major groove is exposed[9,10]. These findings suggest that twisting of DNA may provide a hard constraint rather than an elastic one, on the integration process. Second, DNA supercoiling can transform local twist deformations into global

writhing, i.e. into non-local crossings of DNA's centre axis[30]. This process, eventually culminating in the creation of plectonemes, globally increases the levels of bending along the supercoiled molecule. In line with this, it has been shown that integration along supercoiled DNA is ~2–5-fold enhanced with respect to relaxed DNA[31]. Based on our previous results, we would predict that, in addition of being enhanced in supercoiled DNA, integration should be favoured at the tips of plectonemes, as these display the largest bending compared with the rest of the higher-order structure. We highlight that this argument shares the same principles recently employed to explain why plectonemes preferably appear, and are pinned at, DNA mismatches or weak DNA bending spots[32,33]. In turn, this analogy can be used to extend our argument to predict favoured integration within DNA defects or kinks. We hope that these predictions may be tested in future experiments.

**Local chromatin structure affects integration**. The results from the previous section point to an intriguing role of DNA elasticity in determining the observed preferred integration within mono-nucleosomes. Yet, poly-nucleosome (or chromatin) fibre is a more natural substrate for retroviral integration in vivo. To address this level of detail we now model a 290 bead (~2.1 kbp) long chromatin fibre, forming an array of 10 nucleosomes.

The self-assembly of the fibre is guided by the same principles of the previous section (see also Supplementary Note 6). Attractive interactions ($\epsilon = 4k_BT$) are assigned between nucleosomal DNA (20 beads or ~147 bp) and histone-like proteins (size $\sigma_h = 3\sigma = 7.5$ nm). Linker DNA (10 beads or ~74 bp) separates 10 blocks of nucleosomal DNA and the stiffness of the DNA is fixed at $l_p = 20\sigma = 50$ nm. The ground state of this model is an open chromatin fibre, similar to the 10-nm fibre (Fig. 2a). [While the size of our linker DNA is slightly above the average one in eukaryotes, this is chosen to accelerate the self-assembly kinetics of the fibre as the energy paid to bend the linker DNA is lower[34]] To generate increasing levels of compaction, thus mimicking different local chromatin environments, we introduce an affinity (or attraction) between selected histone-like proteins. We consider either the case where each nucleosome, labelled $i$, interacts with its nearest neighbours (nn) along the chain, $i \pm 1$, or with its next-to-nearest (nnn) neighbours, $i \pm 2$. The former case leads to bent/looped linker DNA[34,35] while the latter a local zig–zag folding displaying straight linker DNA[36] (Fig. 2b–d). Importantly, recent evidence from both in vitro[37] and in vivo[38,39] suggest that both these types of conformations may occur in different regions of the genome—so that the associated chromatin fibre is 'heteromorphic'[34,40,41]. For the nearest-neighbour case, we also distinguish a partially folded state (nnp)—obtained when $\epsilon_h = 40k_BT$ (Fig. 2b)—and a fully condensed structure (nnf)—when $\epsilon_h = 80k_BT$ (Fig. 2c).

By simulating quasi-equilibrium stochastic retroviral integration within these chromatin fibres, we observe that their folding yields a notable effect. Whilst open fibres still display a preference towards nucleosomes that is significantly enhanced with respect to random distributions, this bias is weakened for more compact fibres, especially for nearest-neighbour folding (Fig. 2e, g). What underlies this change in trend? We analyse the local bending energy landscape along the polymer contour and discover that nearest-neighbour (nnp and nnf) folding requires looping of the short linker DNA (Fig. 2f). In turn, this forced looping increases the local bending stress in the linker DNA, potentially lowering the energy barrier against integration comparable to the one stored within histone-bound regions (Fig. 2e).

Whilst this argument could explain why linker and nucleosomal DNA may become equally targeted, it fails to explain why

linker DNA is favoured over nucleosomal one in highly compacted chromatin fibres (nnf, Fig. 2c). It also fails to explain the decreased preference for nucleosomal DNA in zig–zag chromatin fibres, where the linker DNA is straight (nnn, Fig. 2d). We therefore reason that a second important factor is dynamic accessibility: when nucleosomes are tightly packed against each other, there is less available 3D space to reach them diffusively thus hindering integration efficiency. This is true especially for the highly condensed structure with nearest-neighbour (nnf) attraction, which indeed leads to the most striking reduction in nucleosomal integration (Fig. 2g).

Another testable result of our simulations is that the overall integration efficiency, measured by number of integrations $n_{int}$ over the total simulation time, is reduced by chromatin compaction, and integration in a zig–zag fibre yields the globally slowest process. This suggests that integration may be more efficient in vivo within open structures, such as euchromatin, with respect to compact ones, normally associated with heterochromatin.

Although a generic tendency of HIV integration to be suppressed in compacted chromatin has been shown in the past[11,13,42], no existing experiment has accurately measured retroviral integration profiles within chromatin fibres displaying different folding patterns. Thus, we hope that our predictions may be tested in the future by considering reconstituted chromatin in vitro at different salt concentrations.

**Chromatin accessibility favours integration in euchromatin**. Polymer modelling of large-scale 3D chromatin organisation has greatly improved our current understanding of genome architecture in vivo[23,24,43–48]. Some of these models strongly suggest that epigenetic patterns made of histone post-translational modifications—such as H3K4me3 or H3K9me3—play a crucial role in folding the genome[44,47,48]. Based on this evidence we thus ask whether a polymer model of viral integration on a chromatin fibre that is folded in 3D based upon its epigenetic patterns can give us some insight of how large-scale 3D chromatin architecture determines the distribution of integration sites.

To do so, we coarse-grain a poly-nucleosomal fibre in a polymer of thickness $\sigma = 10$ nm (about the size of a nucleosome). We further assume that the histones carry epigenetic marks which drive the large-scale chromatin folding. We then perform our quasi-equilibrium sotchastic integration process within these pre-folded substrates. We start from an idealised block co-polymer model in which 50 blocks of 100 beads are portioned into 30 euchromatic and 70 heterochromatic beads (fraction of heterochromatin $\phi_{het} = 70\%$, see Fig. 3). Heterochromatic compaction is driven by implicit multivalent bridges[49,50], which are effectively accounted for by endowing heterochromatic beads with a weak self-attraction ($\epsilon = 3 k_BT$, see refs. [44,47]). In contrast, we assume that euchromatic beads interact only by steric repulsion, for simplicity. This model naturally drives the phase-separation of the system into compartments of compact heterochromatin (or 'B') decorated by swollen loops of euchromatin (or 'A') as seen in experiments[27]. Strikingly, we observe that the integration probability is highly enriched in euchromatic regions (Fig. 3c), even by setting the same persistence length everywhere along the fibre ($l_p = 3\sigma$). Thus, we infer that large-sacle chromatin folding provides a second driver, besides flexibility, favouring retroviral integration in active region, in excellent qualitative agreement with experiments[1,15,16].

Our simulations give a mechanistic insight into the biophysical mechanism which may underlie this phenomenon. Inspection of the simulations trajectories suggests that a recurrent structure in our model is a daisy-like configuration (see Fig. 3a), with one or

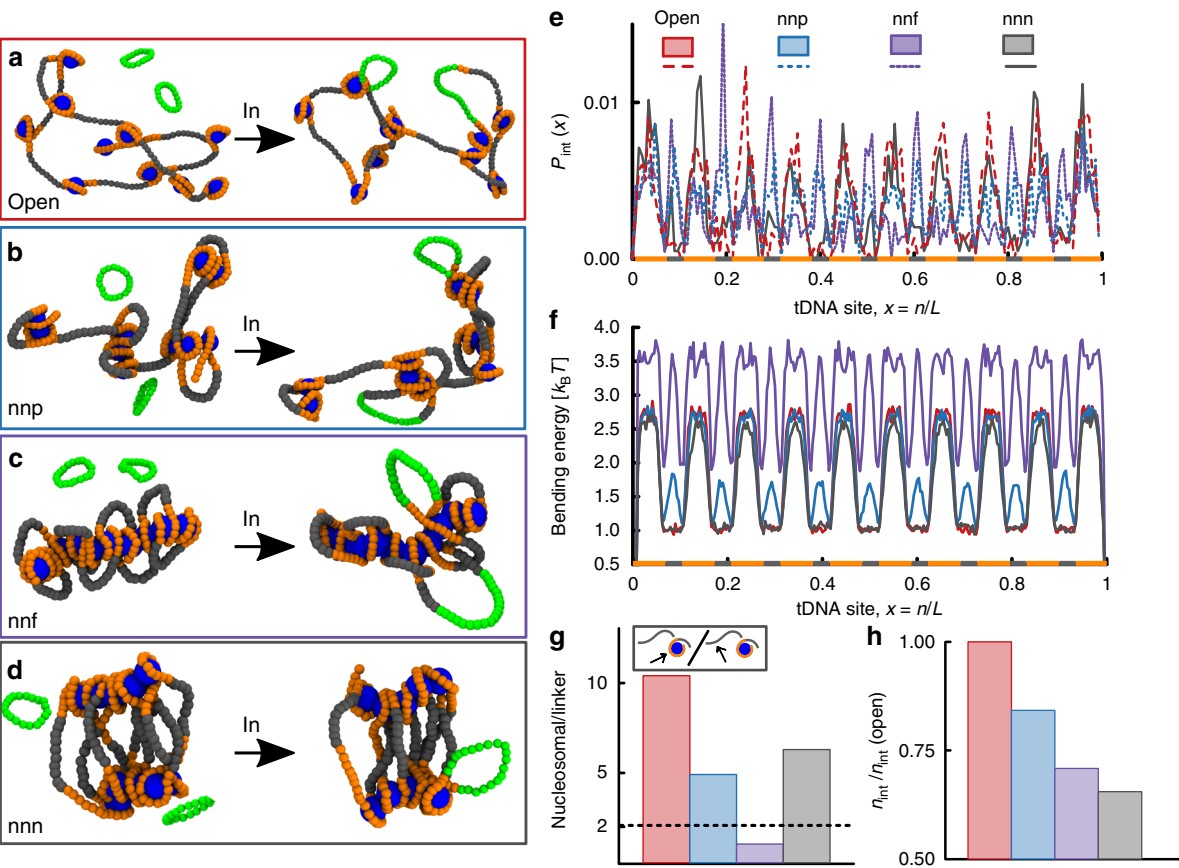

**Fig. 2** Local chromatin structure affects nucleosomal integration. **a** Snapshot of an open chromatin fibre composed of 10 nucleosomes. Nearest-neighbour (nn) attraction between the histone-like particles induce partially folded (nnp, (**b**)) and fully condensed (nnf, (**c**)) structures. Next-nearest-neighbour attractions lead to zig–zagging fibres (nnn, (**d**)). **e** The integration probability $P_{int}(x)$ as a function of the relative tDNA site $x = n/L$ displays peaks whose location depend on the compaction level. Open fibres are integrated mostly within nucleosomes while folded arrays also display peaks within linker DNA. **f** The bending energy profile shows that fibres with nearest-neighbour attractions (nnp and nnf) but not those with next-nearest-neighbour attraction, display stress within linker DNA. This only partially explains why these regions are targeted within these chromatin structures. **g** The ratio of nucleosomal versus linker DNA integrations suggest that not only energy barrier but also dynamic accessibility plays a role in determining the integration profiles (the expected value for random integration $200/90 = 2.2$ is shown as a dashed line). **h** The number of successful integration events over the total simulation time, $n_{int}$, decreases with chromatin condensation. In all cases, the fibre is reconstituted independently before performing the quasi-equilibrium stochastic integration. Data are generated by averaging over 2000 independent integration events. See Supplementary Movies 3–6 for the full dynamics. Source data are provided as a Source Data file

more large heterochromatic cores, screened by many euchromatic loops (petals). The latter are therefore the regions which first encounter the diffusing vDNA; a similar organisation is expected near nuclear pores of interphase nuclei, where channels of low density chromatin separate inactive and lamin-associated regions of the genome[51].

**Retroviral integration profiles in human chromosomes**. To quantitatively test our generic co-polymer model for inter-phase chromosomes, we consider a region of the chromosome 11 in Jurkat T-cell (46–51 Mbp). We coarse-grain the chromatin fibre into beads of size $\sigma = 1\,kbp \simeq 10\,nm$, and label them as euchromatin (red) if the corresponding genomic location simultaneously display high GC content and high expression in the Jurkat cell line (see Fig. 3d, data available from ENCODE[52] and ref. [16]). The remaining beads are marked as heterochromatin (blue). The threshold in GC content and expression level is set in such a way that the overall heterochromatin content is ~70%. We then compare the statistics of integration events that occur within a folded chromatin fibre (by imposing a weak heterochromatin self-attraction, as before $\epsilon = 3\,k_BT$) and within a non-folded substrate

(by imposing repulsive interaction between any two beads irrespectively if hetero- or euchromatic).

Our findings confirm that the reason behind the non-random distribution of integration sites within this model is indeed the 3D folding of the chromatin fibre, as we instead find a uniform (random) integration probability in the unfolded case (see Fig. 3e and Supplementary Movie 3). We finally compare the distribution of predicted integration sites with those detected by genome-wide sequencing in ref. [16] (Fig. 3f). We do this by testing the independence of the integration profiles in real T-cells and in silico using a Spearman Rank test. This test reports a highly significant agreement ($p < 0.001$) between experimental and simulated retroviral integration profiles in folded substrates and it confirms that there is no correlation ($p \simeq 0.6$) between experimental profiles and the ones found along unfolded chromatin substrates.

Our results suggest that large-scale 3D chromatin organisation is an important physical driver that can bias the distribution of integration sites even when the substrate displays uniform elasticity. Because of the daisy-like conformation assumed by folded chromosomes—with a heterochromatic core screened by euchromatic 'petals'—the integration events are more likely to

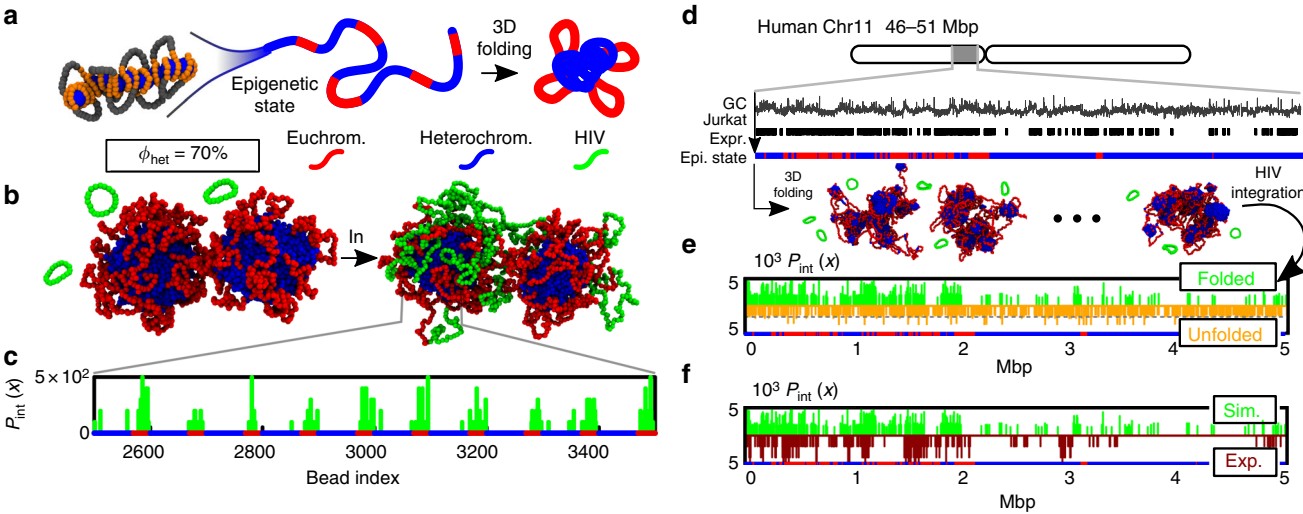

**Fig. 3** Large-scale 3D chromatin folding enhances euchromatic integration. **a** Pictorial representation of our coarse-grained model which describes chromatin as a fibre with epigenetic marks. These marks dictate 3D folding by self-association through proteins and transcription factors[22,49]. **b** Snapshots of our polymer model where the fraction of heterochromatin is set at $\phi_{het} = 70\%$. We show two typical configurations, before and after integration events. **c** The integration probability displays a strong enrichment in euchromatic regions. **d–f** Simulations of a 5 Mbp region of human chromosome 11 (46–51 Mbp) modelled at 1 kb resolution with a polymer $N = 5000$ beads long. **d** In this model, expression level in Jurkat T-cells and GC content are used to label beads as euchromatic (red) or heterochromatic (blue), respectively. We assign attractive interactions ($\epsilon = 3\,k_BT$) between heterochromatin beads so that the fixed epigenetic pattern guides the folding of the chromatin fibre (see snapshots). Steady-state conformations are then used as hosts for $n = 500$ integration events of a 10 kbp viral DNA. **e** Comparison between the distribution of integration sites in folded and unfolded chromatin conformations. The latter is obtained by assigning no self-attraction between heterochromatin beads. Viral integration within unfolded chromatin is uniform ($P_{int} = 1/n$, dashed line) while it is not uniform (i.e. non-random) on folded chromatin. **f** Comparison between simulated and experimentally measured distribution of integration sites in Jurkat T-cells[16]. The agreement between simulations and experiments is highly significant, with a $p$-value $p < 0.001$ when a Spearman Rank is used to test the null-hypothesis that the distributions are independent. This result can be compared with the $p$-value $p = 0.6$ obtained when the same test is performed to test independence of the integration profiles in experiments and unfolded chromatin. The dynamics corresponding to one of our simulations is shown in full in Supplementary Movie 7. Source data are provided as a Source Data file

occur on euchromatin regions as these are the most easily accessible. We thus conclude this section by suggesting that large-scale chromosome folding is a generic physical driver that underlies integration site-selection for all families of retrovirus that target interphase nuclei (this argument therefore excludes alpha- and beta-retroviruses). Specifically, we find that the bias for open chromatin is a direct consequence of diffusive target search along a pre-folded substrate and we argue that this mechanism is at work even in absence of known tethering factors such as LEDGF/p75. Although this nuclear protein is known to enhance the preference of HIV for euchromatin[15], it is also well-established that this preference remains significantly far from random in cells where LEDGF/p75 is knocked-out[15,17,19]. This unexplained discrepancy can be rationalised within our model as originating from purely physical mechanisms.

**Heterochromatin content strongly affects integration.** Distinct cell types may display dramatically different amounts of active and inactive chromatin, and this aspect has been shown to affect HIV integration efficiency, at least in some cases. Most notably, a 'resting' T-cell, which contains a larger abundance of the H3K9me3 mark[53,54] and of cytologically defined hetero-chromatin, has been shown to be less likely to be infected by HIV with respect to an 'activated' T-cell. It is also known that the few resting cells which get infected do so after a sizeable delay[55]. To shed light into these unexplained findings, we now consider a block co-polymer chromatin model with varying fraction of heterochromatin content ($\phi_{het} = 30$, 50 and 80%).

In Fig. 4a, b we show typical 3D structures of chromatin fibres with different heterochromatic content. When the latter is small ($\phi_{het} = 30\%$), heterochromatin self-organises into globular compartments of self-limiting size, surrounded by long euchromatin

loops which entropically hinder the coalescence of heterochromatic globules[56,57]. For large heterochromatin content ($\phi_{het} = 80\%$), inactive domains merge to form a large central core, 'decorated' by short euchromatic loops, resembling the above-mentioned daisy-like structure. Our simulations confirm that viral loops integrate preferentially in open, euchromatin regions in all these cases. Additionally, we observe that the total time taken for the viral loops to integrate within the genome increases (super-)exponentially with the abundance of heterochromatin (Fig. 4c). In biological terms this implies that 'resting', heterochromatin rich, T-cells take much longer to infect with respect to activated T-cells. The same results additionally suggest that stem cells, which are euchromatin-rich, should be infected more quickly with respect to differentiated cells[58]. Both findings are in qualitative agreement with existing experiments on lentivirus infection[55,59].

A further surprising result is that the efficiency of viral integration in the euchromatic parts of the genome increases with the total fraction of heterochromatin. This can be quantified by measuring the integration probability within a given epigenetic state, $s$, as

$$P_{int}^s = \sum_{i=1}^{N} P_{int}(i)\delta(s(i) - s) \qquad (1)$$

where $s(i)$ is the epigenetic state of the $i$-th bead. For random integration events, i.e. constant $P_{int} = 1/N$, one obtains $P_{random}^s = \phi_s$. Hence the change in integration efficiency due to the 3D organisation can be quantified as $\chi_s = P_{int}^s/\phi_s$ (see Fig. 4d).

We find that $\chi_{eu}$ increases with $\phi_{het}$, while $\chi_{het}$ decreases. This counterintuitive observation can be understood as a

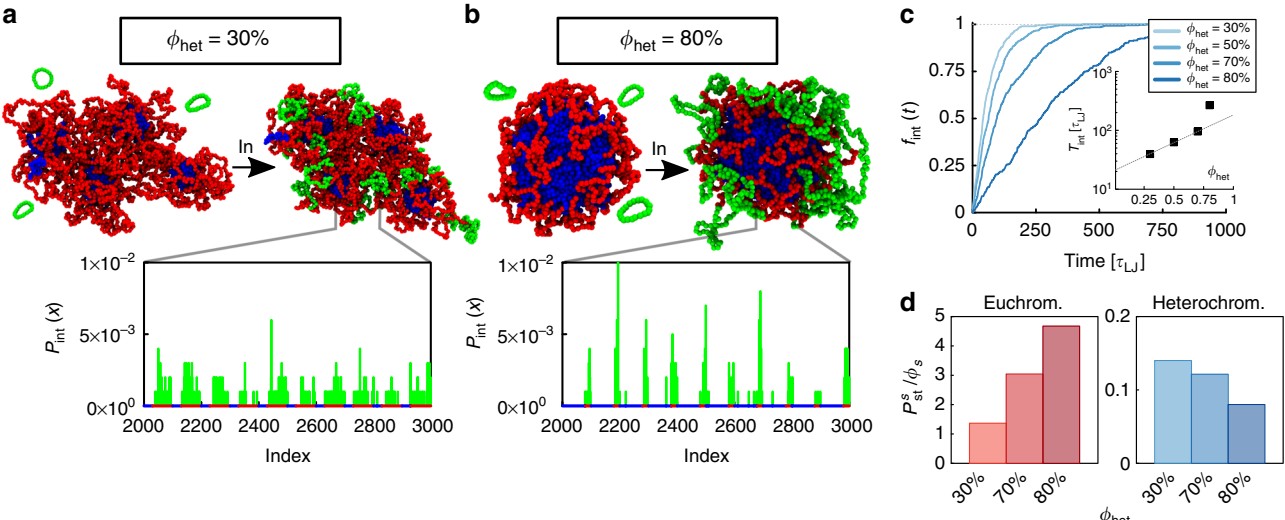

**Fig. 4** Integration is slowed down in cells with large heterochromatin content. **a, b** Snapshots and probability distribution for a system with $\phi_{het} = 30$ and 80%, respectively. **c** Fraction of integrated loops $f_{int}$ as a function of time, for different levels of heterochromatin. In the inset, the integration time $T_{int}$ defined as $f_{int}(T_{int}) = 0.5$ is shown to (super-)exponentially increase as a function of $\phi_{het}$. **d** Integration probability in state $s$, with $s$ being either euchromatin (red) or heterochromatin (blue), normalised by the fraction of the host polymer in $s$, $\phi_s$. These show that the larger the fraction of heterochromatin, the more likely it is for a virus to be integrated in euchromatin. Source data are provided as a Source Data file

direct consequence of 3D chromatin architecture. The more heterochromatin is present in the nucleus, the stronger the inactive ('B') compartments and the more they are screened by euchromatin loops. As far as we know, this finding has never been directly observed and it would be interesting to test in the future.

**A reaction–diffusion model for nuclear integration**. Having observed that large-scale chromosome folding can affect the distribution of retroviral integration sites trough chromatin accessibility, we now aim to put this finding into the context of a realistic inter-phase nuclear environment. Because performing polymer simulations of a full genome is not currently feasible, we consider the observations made in the previous sections to formulate a continuum model of whole cell nuclei. We do so by coarse graining the behaviour of retroviral DNA in the nucleus as a random walk inside a sphere of radius $R$, which can integrate into the host genome at a rate $\kappa$. In general, the diffusion constant $D$ and the integration rate $\kappa$ will depend on the position of the viral loop in the nuclear environment. Indeed we have seen before that local epigenetic state and chromatin architecture play important roles in determining retroviral integration rate and patterns.

Within this model, the probability $\rho(\boldsymbol{x}, t)$ of finding a viral loop in the nucleus at position $\boldsymbol{x}$ and time $t$ obeys the following reaction–diffusion equation:

$$\partial_t \rho(\boldsymbol{x}, t) = \nabla(D(\boldsymbol{x})\nabla\rho(\boldsymbol{x}, t)) - \kappa(\boldsymbol{x})\rho(\boldsymbol{x}, t). \quad (2)$$

For simplicity, we assume spherical symmetry, i.e. $\rho(r, \theta, \phi, t) = \rho(r, t)$, and piecewise constant functions for $D$ and $\kappa$ (see below). With these assumptions, Eq. (2) becomes $\partial_t \rho = D/r^2 \partial_r (r^2 \partial_r \rho) - \kappa\rho$, where we have dropped, for notational simplicity, all dependences on $r$ and $t$. In order to obtain the steady-state probability of integration sites, we thus need to find the time-independent distribution $\rho(r, t) = \rho(r)$ by solving the equation

$$\frac{D}{r^2}\partial_r (r^2 \partial_r \rho) - \kappa\rho = 0. \quad (3)$$

In the simplest case in which $D$ and $\kappa$ are uniform throughout the

nucleus, the solution of Eq. (3) is

$$\rho(r) = \mathcal{N}\frac{\sinh(r/l)}{r}, \quad (4)$$

where $l = \sqrt{D/\kappa}$ is a 'penetration length', measuring the typical lengthscale that vDNA diffuses before integrating into the tDNA, while $\mathcal{N}^{-1} = \int_0^R dt \sinh(t)/t$ is a normalisation constant (see Supplementary Note 8). To solve Eq. (2) in more general cases, we need to make some assumptions on how $D$ and $\kappa$ may vary within the nuclear environment. In line with our previous results at smaller scale, we assume that these parameters depend on local chromatin state—as we shall discuss, this is often dependent on nuclear location.

First, we need to model viral diffusivity in euchromatin and heterochromatin. Assuming faster or slower diffusion in euchromatin are both potentially reasonable choices: the former assumption describes situations where euchromatin is more open and less compact[60,61], whereas slower diffusivity may instead model local gel formation by mesh-forming architectural proteins such as SAF-A[62]. Second, the recombination rate $\kappa$ may be thought of as effectively depending on local DNA/chromatin flexibility and 3D conformation, and given our previous results we expect it to be larger in euchromatin-rich nuclear regions. For simplicity, we additionally posit that chromatin is organised in the nucleus into three main concentric zones. Each of these zones displays an enrichment of a particular chromatin state. This is the situation of typical differentiated cells, where it is well-established that the most inner and outer zones are generally populated by transcriptionally inactive chromatin (heterochromatin and lamin-associated-domains, respectively[2,63]), whereas the middle layer is commonly enriched in transcriptionally active euchromatin[64]. To mimic this organisation, and for simplicity, in our model $D$ and $\kappa$ vary spatially, but we assume a constant value within each of the three layers. Whilst this may be a crude approximation for individual cells, which are known to display heterogeneity in the local chromosome organisation[65], our model may be more suitable to capture behaviours from population averages (see Fig. 5).

To highlight the effect of nuclear organisation on the spatial distribution of retroviral integration sites, we compare the case

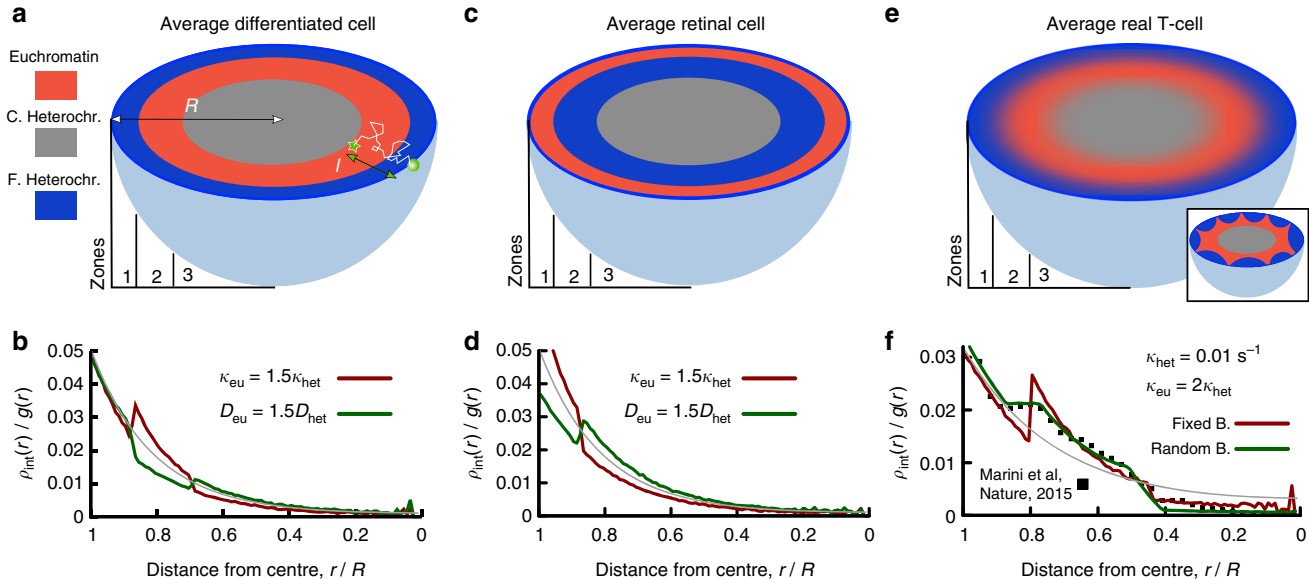

**Fig. 5** HIV integration hot-spots are affected by nuclear organisation. **a, c, e** Different cell lines display different chromatin organisations at the nuclear scale. **a** Shows a typical differentiated cells, modelled as a sphere with 3 concentric shells of equal volume. Zones 1 and 3 are populated by facultative and constitutive heterochromatin, respectively. Zone 2, the middle layer, is populated by euchromatin. This configuration may be viewed as an angularly averaged model and it is appropriate to study HIV integration in population averages. **c** Shows the model for a 'retinal' cell, where the two outer layers are inverted[64]. **e** Shows the model for a realistic population of T-cells (typical configuration of a single cell is shown in the inset). Here the location of the boundaries between zones 1 and 2, and between zones 2 and 3, is varied to account for local density variations and cell-to-cell fluctuations (see text). **b, d, f** Nuclear distribution of HIV integration sites in **b** differentiated cells, **d** retinal cells and **f** T-cells. The result with uniform $D$ and $\kappa$ (yielding $l \simeq 2.23\,\mu m$) is shown in grey in each panel. The number of integrations at distance $r$, $\rho_{int}(r)$, is divided by the area of the shell, $g(r) = 4\pi r^2 dr$, and normalised to unity. Filled squares in **f** denote data from ref. [66]. Source data are provided as a Source Data file

just discussed of a differentiated cells, displaying a conventional layering, with cells displaying an 'inverted' organisation, such as the retinal cells of nocturnal animals[64] (see Fig. 5c). By measuring the integration profile $\rho_{int}(r)$ (normalised by the area of the shell $g(r) = 4\pi r^2 dr$), we how the integration profile changes because of non-uniform $D$ and $\kappa$ (see Fig. 5b, d). As expected, we find that setting the recombination rate in euchromatin, $\kappa_{eu}$, larger than the one in heterochromatin, $\kappa_{het}$, enhances the probability of integration in the middle euchromatic layer in differentiated cells. On the other hand, faster diffusion in euchromatin-rich regions has the opposite effect (as fast diffusion depletes virus concentration). We also predict that the distribution of integration events in retinal cells should be very different. Here, larger $\kappa_{eu}$ enhances the probability of integration near the periphery and, as before, increasing $D_{eu}$ has the opposite effect.

**A refined model explains observed HIV hot-spots in T-cells.** We now quantitatively compare our reaction–diffusion model with the experimentally measured distribution of HIV recurrent integration genes (RIGs) in T-cells[66]. Remarkably, we find that our simple theory with uniform $D = 0.05\,\mu m^2/s$ and $\kappa = 0.01\,s^{-1}$ (leading to a penetration length $l = \sqrt{D/\kappa} = 2.23\,\mu m$) is already in fair overall agreement with the experimental curve (grey line in Fig. 5f). The agreement can be improved by progressively refining our model and adding more stringent assumptions. As we show below, these refinements also lead us to obtain more physical insight into the nuclear organisation of chromatin in real T-cells.

First, we find that a better agreement is achieved if $D$ remains uniform, but $\kappa$ varies and $\kappa_{eu}/\kappa_{het} \simeq 1.5$ (equivalently, a model with $D_{het}/D_{eu} \simeq 1.5$ and uniform $\kappa$ would yield the same result). A second improvement is found by re-sizing the three concentric nuclear shells as follows. We maintain the total mass of heterochromatin fixed at twice that of euchromatin, as realistic in vivo[2]. The volume of each layer has to adapt according to the

fact that active chromatin is less dense than heterochromatin[51,60]. It is possible to derive an equation relating the ratio between the density of heterochromatin and euchromatin, $\rho_{het}/\rho_{eu}$, to the positions of the boundaries between layers, $R_{1-2}$ and $R_{2-3}$, as (see Supplementary Note 10)

$$\frac{\rho_{het}}{\rho_{eu}} = \frac{2\left(R_{2-3}^3 - R_{1-2}^3\right)}{R^3 + R_{1-2}^3 - R_{2-3}^3}. \tag{5}$$

For a nucleus of radius $R$, we find that setting $R_{2-3}/R = 0.8$ and $R_{1-2}/R = 0.445$ (to match the data from ref. [66]) we obtain $\rho_{het}/\rho_{eu} = 1.6$. This value is in pleasing agreement with recent microscopy measurements, reporting a value of $1.53$[67].

Two final refinements that we consider here are allowing for small fluctuations (up to a maximum of $\pm 0.5\,\mu m$) in the position of the boundaries in each simulated nucleus and imposing that the innermost boundary, between euchromatin and constitutive heterochromatin, can be crossed by the viral loop only with probability $p = 0.2$. The former assumption accounts for the heterogeneity in a population of cells (as the positions of the boundaries between zones are not fixed) while the latter is related to the exclusion from the nucleolus. By including these realistic refinements, our theory matches extremely well the experimental measurements (Fig. 5f). [We provide a more quantitative estimation on the goodness of our model with respect to these parameters in Supplementary Fig. 3 and Supplementary Note 11].

**Applications to HIV infection.** In this work we have proposed a generic physical model for retroviral integration in DNA and chromatin, which is based on 3D diffusive target search and quasi-equilibrium stochastic integration. Importantly, our model purposely neglects the interaction of the PIC with other co-factors and nuclear pore proteins[15,17,68]. We make this choice both for simplicity and to focus on the key physical ingredients that are

necessary and sufficient to recapitulate a bias in the site-selection process but have been overlooked in the past.

Our model can be directly applied to the case of HIV infection. In this case, it is known that the interaction of HIV with nucleoporins and cellular proteins is mainly relevant to ensure its successful nuclear entry[3,20]. Similarly, the cleavage and poly-adenylation specificity factor 6 (CPSF6) is known to bind to the viral capsid, and is required for nuclear entry[3]. Recent evidence suggest that CPSF6 may also contribute to the integration site-selection[68]; yet, while the viral capsid is present in the nucleus of primary macrophages[69], there is no evidence suggesting that this enters the nucleus of primary T-cells, thus questioning the relevance of CPSF6 in this cell lineage (on which we focus when comparing to human chromosome HIV integration patterns).

Finally, while it is well-established that the presence of functional LEDGF/p75 enhances euchromatic HIV integration[3,15], this preference is found to persist significantly above random when this co-factor is knocked-out[15,17,19]. All this calls for a model that can explain non-random HIV site-selection independently of other co-factors, such as the one we have proposed here.

In the future, it would be possible to consider a refinement of our model in which a euchromatin tethering factor such as LEDGF/p75 is accounted for by setting specific attractive interactions between the vDNA polymer and euchromatic regions. This refined model is expected to naturally result in an enhancement of euchromatic integrations since the vDNA would spend more time in their vicinity. While we here find that this element is not necessary to recapitulate the preference for euchromatin, we realise that it may be interesting to study its role within the context of a more realistic interphase nuclear environment, where the virus has to traffic through a complex and crowded space. We leave this investigation for subsequent studies.

We finally argue that because our model is based on few generic assumptions, i.e. that of diffusive search and energy barrier sensing, our results are expected to hold for a number of retroviral families as long as their members undergo diffusion within an interphase nucleus and require bending of the tDNA substrate to perform integration. Important exceptions are the families of alpha- and beta-retroviruses as they possess a unique intasome structure which may accommodate unbent tDNA[70,71] and can only infect mitotic cells[15].

## Discussion

In this work we propose a generic biophysical model to rationalise the problem of how some families of retroviruses, and in particular HIV, can display non-random distributions of integration sites along the genome of the host. Our model identifies two key physical features underlying this non-trivial selection: local genome elasticity and large-scale chromatin accessibility. These two biophysical drivers are active at multiple length scales, and create trends which are in qualitative and quantitative agreement with experimental observations. Importantly, we stress that these two mechanisms are at play even in absence of known co-factors, for instance in vitro or in knock-out experiments[8,15,19], and should thus be considered as forming the physical basis of retroviral integration.

By modelling integration events as stochastic and quasi-equilibrium topological reconnections between 3D-proximal polymer segments we find a bias towards highly bent or flexible regions of the genome, in quantitative agreement with long-standing experimental observations (Fig. 1). This bias can be explained as resulting from the difference in energy barrier against local deformation of the underlying tDNA substrate.

Because highly bent, nucleosomal DNA is associated with a low energy barrier and thus geometrically catalyses integration.

At intermediate scales, we find that a poly-nucleosomal chromatin fibre can display a wide range of different integration patterns depending on the level and type of folding. Notably, solenoidal fibres would give a marked increase in integration in linker DNA at the expense of nucleosomal one, due to DNA accessibility (Fig. 2). We argue that integration patterns on chromatin templates in vitro may thus shed light into their local structure, an open question in chromatin biology.

At larger scales, our model predicts retroviral integration patterns closely matching experimental ones and showing a marked preference towards transcriptionally active euchromatin (Fig. 3). This can be explained by noting that 3D chromosome folding, dictated by the underlying epigenetic marks, determines chromatin accessibility (Figs. 3 and 4). In general, we argue that any vDNA that probes space diffusively must be affected by large-scale 3D folding of the substrate. In line with this, we further predict that cell lines displaying a large abundance of hetechromatin should be infected by HIV less efficiently than ones richer in euchromatin (Fig. 4). Finally, we propose and solve a simple reaction–diffusion model that can capture the distribution of integration hot-spots within the nuclear environment in human T-cells (Fig. 5).

Besides rationalising existing evidence on retroviral integration by using minimal assumptions, our model leads to a number of testable predictions. For instance, we find that integration events that are not in 'quasi-equilibrium', i.e. that consume ATP to deform the substrate, cannot sense regions of lower energy barrier and thus do not display any bias towards nucleosomal or flexible DNA. This scenario may be relevant if intasome complexes expending ATP are found, or artificially built. At the chromosome level, our model can be used to predict the distribution of integration sites within chromosomes with known epigenetic patterns. Thus, it can potentially be used to predict generic, i.e. not co-factor specific, retroviral integration profiles in a number of different cell lineages and organisms. These results could finally be compared, or combined, with 'chromosome conformation capture', e.g. HiC, analysis to provide valuable insight into the relationship between retroviral integration and large-scale chromatin organisation in living cells[27,66]. At the whole cell level, our reaction–diffusion model can be used to predict how the distribution of HIV hot-spots may change in cells with non-standard genomic arrangements, such as retinal cells in nocturnal animals[64], but also senescent[72] and diseased cells in humans and mammals.

## Methods

**Computational details**. To model DNA and chromatin, we consider a broadly employed coarse-grained model for biopolymers[24,25,47,49]. In this model, DNA and chromatin are treated as semi-flexible bead-spring chains made of $M$ beads. Each bead has a diameter of $\sigma$, which is taken to be $\sigma = 2.5$ nm (or 7.35 bp) for DNA and $\sigma = 10$ nm (or 1 kbp) for chromatin. We simulate the dynamics of the fibre by performing MD simulations in Brownian scheme, i.e. we include a stochastic force on each monomer to implicitly account for the solvent and noisy environment. As commonly done in MD simulations, we express properties of the system in multiples of fundamental quantities. Energies are expressed in units of $k_B T$, where $k_B$ is the Boltzmann constant and $T$ is the temperature of the solvent. Distances are expressed in units of $\sigma$, which, as defined above, is the diameter of the bead. Time is expressed in units of the Brownian time $\tau_{Br}$, which is the typical time for a bead to diffuse a distance of its size, more precisely, $\tau_{Br} = \sigma^2/D = 3\pi\eta\sigma^3/k_B T$, where $D$ is the diffusion constant for a bead, and $\eta$ the solvent viscosity.

The interactions between the beads are governed by several potentials that are standard in polymer physics. First, purely repulsive interactions are modelled by the standard Weeks–Chandler–Anderson potential

$$U_{WCA}^{ab}(r) = k_B T \left[ 4 \left[ \left( \frac{\sigma}{r} \right)^{12} - \left( \frac{\sigma}{r} \right)^6 \right] + 1 \right] \qquad (6)$$

if $r < r_c = 2^{1/6}\sigma$ and 0 otherwise. Here, $r$ is the separation between the two beads

and $r_c$ is a typical cut-off to ensure that the interaction is repulsive. Second, bonds between consecutive beads are treated as finitely extensible (FENE) springs:

$$U_{\mathrm{FENE}}^{ab}(r) = -\frac{K_f R_0^2}{2}\ln\left[1 - \left(\frac{r}{R_0}\right)^2\right]\left(\delta_{b,a+1} + \delta_{b,a-1}\right), \qquad (7)$$

where $R_0$ (set to 1.6$\sigma$) is the maximum separation between beads and $K_f$ (set to $30k_BT/\sigma^2$) is the strength of the spring. The combination of the WCA and FENE potentials with the chosen parameters gives a bond length that is approximately equal to $\sigma$[49]. Third, we model the stiffness of the polymers via a Kratky–Porod term:

$$U_{\mathrm{KP}}^{ab} = \frac{k_B T l_p}{\sigma}\left[1 - \frac{\boldsymbol{t}_a \cdot \boldsymbol{t}_b}{|\boldsymbol{t}_a||\boldsymbol{t}_b|}\right]\left(\delta_{b,a+1} + \delta_{b,a-1}\right), \qquad (8)$$

where $\boldsymbol{t}_a$ and $\boldsymbol{t}_b$ are the tangent vectors connecting bead $a$ to $a+1$ and $b$ to $b+1$, respectively; $l_p$ is the persistence length of the chain and is set to $l_p = 20\sigma = 50$ nm for DNA and to $l_p = 3\sigma \approx 30$ nm for chromatin[27].

When needed, attractive interactions are modelled via a standard Lennard–Jones potential

$$U_{\mathrm{LJ}}^{ab}(r) = 4\epsilon\left[\left(\frac{\sigma}{r}\right)^{12} - \left(\frac{\sigma}{r}\right)^6 - \left(\frac{\sigma}{r_c}\right)^{12} + \left(\frac{\sigma}{r_c}\right)^6\right] \qquad (9)$$

if $r \le r_c = 1.8\sigma$ and 0 otherwise.

To summarise, the total potential energy related to bead $a$ is the sum of all the pairwise and triplet potentials involving the bead:

$$U_a = \sum_{b\neq a}\left(U_{\mathrm{WCA}}^{ab} + U_{\mathrm{FENE}}^{ab} + U_{\mathrm{KP}}^{ab} + U_{\mathrm{LJ}}^{ab}\right). \qquad (10)$$

The time evolution of each bead along the fibre is governed by a Brownian dynamics scheme with the following Langevin equation:

$$m_a\frac{d^2\boldsymbol{r}_a}{dt^2} = -\nabla U_a - \gamma_a\frac{d\boldsymbol{r}_a}{dt} + \sqrt{2k_BT\gamma_a}\,\boldsymbol{\eta}_a(t), \qquad (11)$$

where $m_a$ and $\gamma_a$ are the mass and the friction coefficient of bead $a$, and $\boldsymbol{\eta}_a$ is its stochastic noise vector obeying the following statistical averages:

$$\langle\boldsymbol{\eta}(t)\rangle = 0; \quad \left\langle \eta_{a,\alpha}(t)\eta_{b,\beta}(t')\right\rangle = \delta_{ab}\delta_{\alpha\beta}\delta(t - t'), \qquad (12)$$

where the Latin indices represent particle indices and the Greek indices represent Cartesian components. The last term of Eq. (11) represents the random collisions caused by the solvent particles. For simplicity, we assume all beads have the same mass and friction coefficient (i.e. $m_a = m$ and $\gamma_a = \gamma$). We also set $m = \gamma = k_B = T = 1$. The Langevin equation is integrated using the standard velocity–Verlet integration algorithm, which is performed using the Large-scale Atomic/Molecular Massively Parallel Simulator (LAMMPS)[73]. We set the integration time step to be $\Delta t = 0.001\tau_{\mathrm{Br}}$, where $\tau_{\mathrm{Br}}$ is the Brownian time as mentioned previously.

The recombination moves are performed using an in-house modified versions of the 'double-bridging' algorithm implemented in LAMMPS as `fix bond/swap` (see ref. [74] for extensive description). The implemented modifications are tailored to our specific model, i.e. they allow us to perform recombination moves between viral and host polymers (inter-chain reconnections) while avoiding intra-chain (or 'self') reconnections. We also modified this code to perform polymer reconnections that bypass the Metropolis test thus allowing non-equilibrium integration (see Supplementary Note 5). The original code is part of the LAMMPS package[73] (https://lammps.sandia.gov) and the modified versions are freely available and can be requested directly from one of the authors (see Code availability section). The recombination moves are attempted at every time step and between beads that are at most $R_c = 2\sigma$ apart. More details on the recombination algorithm and the analytical solutions of the reaction–diffusion equation are given in Supplementary Notes 1 and 7–10. Supplementary Note 4 contains additional results on the integration within DNA with heterogeneous flexibility, while Supplementary Note 5 discusses the case of non-equilibrium integration.

**Code availability**. The code used for the simulation is LAMMPS, which is publicly available at https://lammps.sandia.gov/. In-house codes written to simulate viral integration as stochastic polymer reconnection events are available from the corresponding author upon request.

**Reporting summary**. Further information on experimental design is available in the Nature Research Reporting Summary linked to this article.

## Data availability

The data that support the findings of this study are available from the corresponding author upon request. The source data underlying Figs. 1–5 are provided as a Source Data file. A reporting summary for this Article is available as a Supplementary Information file.

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

## Acknowledgements

We thank D.W. Sumners and J. Allan for inspiring discussions. This work was supported by ERC (CoG 648050, THREEDCELLPHYSICS). The authors would also like to acknowledge the contribution and networking support by the 'European Topology Interdisciplinary Action' (EUTOPIA) CA17139.

## Author contributions

D.Mi, M.L., D.Ma. and E.O. designed research. D.Mi. and E.O. performed simulations. D.Mi, M.L., D.Ma. and E.O. helped to analyse the results and write the manuscript.

## Additional information

**Competing interests:** The authors declare no competing interests.

