## [Peer Review File · Nature Communications]

Reviewer #1 (Remarks to the Author):

In this manuscript, the authors have studied how HIV-1 integration can be regulated by physical parameters of the nucleus, at different scales of its organization. Using modeling tools, they test how specific physical properties of the cellular DNA and chromatin (such as its elasticity, flexibility or the attraction between distinct chromatin regions) affect the selectivity of integration. They also use a reaction-diffusion model to study the localization of the provirus in the nucleus. Their modeling approaches allow the authors to reproduce physiological specificities of HIV integration sites such as their enrichment in DNA regions covered by nucleosomes, in flexible genomic regions, in euchromatin versus heterochromatin, close to integration sites mapped in infected cells and in euchromatin regions of the nucleus.

This is an original and interesting study, with both fundamental and therapeutic impacts. However, I have three major reserves about this manuscript.

- First, most of the predictions have been made without taking into account (and even without citing corresponding studies) several cellular and viral parameters (like LEDGF/p75, Cpsf6 and specific nuclear pore proteins) that have been demonstrated to be important for the selectivity of HIV integration. Some of these proteins can modulate this process at the nucleosome level and other ones at the level of nuclear organization. The authors could at least have discussed the relevance of their results with regards of several virology studies on this process.

- Second, in the initial model of integration, the viral DNA is mimicked by a circular DNA but no precision is given about its length, its covering by proteins and the presence of the intasome which connects the LTR ends. In structures of this complex (solved by crystal or cryo-em approaches), the two LTR ends form an acute angle and the target DNA is also highly bent. These parameters should be taken into account or at least mentioned to explain how they could affect the selected modeling parameters.

- Third, if the selected physical parameters are sufficient to reproduce major specificities of HIV integration, why are these specificities different between HIV and other retroviruses?

For these reasons, but also because the authors have used very precise physical models, I propose to submit this manuscript to a more specialized journal. I also suggest to insert a more detailed discussion on the biological relevance of the selected modeling parameters and the pertinence of the obtained results with regards of previous studies published in the field of HIV integration.

Reviewer #2 (Remarks to the Author):

I think this manuscript discusses a very nice and novel idea on the mechanisms of viral DNA integration in the human genome. The physics model here introduced is very interesting and well explained. Its implementation is technically demanding, but well tackled by the authors. The results are very interesting, especially considering the simplicity of the considered model. They are in good overall agreement with available experimental data. As a minor comment, I would advise the authors to represent even better the framework of their study and the corresponding literature. In brief, I definitely support this manuscript for publication.

Reviewer #3 (Remarks to the Author):

In their manuscript, entitled “Multi-scale Model Reveals Physical Principles of HIV Integration in Human Nuclei”, Michieletto and colleagues propose a theoretical modeling of HIV integration at different scales, from DNA and nucleosomes to the nuclear 3D organization. This article constitutes to my knowledge one of the first attempt to investigate HIV integration with simple theoretical framework. But as mentioned in my comments listed below, I am not fully convinced by their models (too simplistic) or by the relevance of their investigation, except for the very large scale. The models are not sufficiently well explained and their parametrization not enough justified. Actually there is almost no cross-talk between the scales: it is rather a “multiple-scale” approach than “multi-scale”. In addition there is no model of integration at the scale of the nucleosomal array and chromatin fiber which is, to my opinion, the most relevant scale to investigate. There are many points that have to be clarified. This work is actually too ambitious to provide satisfactory results at all scales; I would suggest to investigate the integration process scale by scale (in independent papers) but more deeply and exhaustively. For these reasons and the concerns listed below I cannot recommend this article for publication in Nature Communication.

Main concerns:

“DNA Elasticity Biases HIV Integration”

1) The model of DNA integration is not clearly explained enough. In particular, the authors should describe more precisely the elastic constraint(s) (bending -and twisting ?-) imposed to the viral and host DNA during the insertion process. I agree that such constraints coming from the local DNA conformation (the viral/host DNAs or/and the final host+viral DNA ?) imposed in the intasome by the integrase complex may provide an indirect read-out mechanism for intasome targeting and viral DNA insertion. But, in their model and numerical simulation, what is the exact bending constraint and thus the value of the bending cost that controls final insertion ? Is it the local bending at the two junctions of the final ligated DNA ? Actually, I thought (see paper Michel F, Crucifix C, Granger F, Eiler S, Mouscadet JF, Korolev S, et al. Structural basis for HIV-1 DNA integration in the human genome, role of the LEDGF/P75 cofactor. The EMBO journal. 2009;28(7):980–91.) that the targeting of the intasome at the host DNA was rather driven by the propensity of the host DNA to be bent (providing a conformation favorable for final ligation with viral DNA, conformation imposed by the integrase complex). However it seems to me that the model proposed by the authors does not account for this constraint at this stage of integration but rather for the bending constraint at the ligation stage. I am not sure that it is equivalent. Authors should clarify this point by illustrating, for example, a typical insertion process (the drawing in Fig 1 E,F are not very clear; for example, what are the two depicted integration products, how are they selected etc...?) and showing which step is controlled by a bending energy activation barrier and of which amplitude ? The exact geometry (bending angle) of the constrained DNA should be more clearly illustrated.

Actually, there might exist different logics of insertion by such indirect readout process depending on the integrases+viral DNA complexes architecture. It could be interesting to discuss that point and by this way to justify their specific choice for the imposed constraint.

2) The authors have not considered twist constraints: is there any argument for that ? Depending on the geometry of the viral DNA in the intasome, a twist deformation might also be applied to the host DNA in order to have the proper chemical phasing (in addition to the proper duplex orientation). Twist (and supercoiling) might be also relevant to understand nucleosome selectivity.

3) As mentioned in the section “A Di-Block...”..., DNA is a heteropolymer whose elastical properties depend on the underlying base sequence. There are sequences that are more flexible or intrinsically bent and that, consequently, may also favor viral DNA integration. So there might be also a bias toward specific sequences even for naked DNA by the same indirect read out process. Authors discuss that point in the next section but I would recommend to discuss this sequence effect here.

4) Concerning the preferential integration in the mononucleosome vs naked DNA, indeed the curved configuration imposed by the wrapping around the octamer may favor the final bent configuration inside the intasome (what about twist ?) but is there a good reason to neglect the interaction between the intasome complex and the nucleosome ? There might be a steric hindrance that would rather inhibit integration but reversely, there might be a specific positive interaction between

integrase (or associated cofactors) and the histones, that would promote the targeting at nucleosome and final integration. In their model, preferential integration is at the dyad position: is there any experimental evidence for that? DNA breathing at the entry-exit might also favor integration due to enhanced accessibility. If there are (in vitro) data of integration (mono-nucleosome vs naked DNA) authors should show to what extent their simple model indeed reproduce them.

5) In vivo, the relevant substrate for integration is the nucleosomal array. Even when considering the possibility of integrating in the naked linker DNAs, the presence of neighbouring nucleosomes may lead to a very different situation than integration in a completely naked DNA. Nucleosomal array and the chromatin fiber is, to my point of view, the very relevant scale to investigate: it is the scale of integrase binding and processing. Furthermore, large scale studies would require to have a model of integration at this scale that could be eventually coarse-grained. This is currently not the case (see next section).

“A Di-Block Polymer Model Predicts Preferential Integration into Flexible Genomic Regions”

6) In this section, I actually don't understand why the authors consider both the DNA and the chromatin systems? I think that all could be done with a chromatin model. The main reason is that there is no large portion of naked DNA in vivo, so it might be not so relevant to study integration within naked DNA at large scales. The other reason (see point 7)) is that I do not believe that DNA can experiment phase separation due to flexibility inhomogeneity.

7) Concerning the di-block copolymer model: for DNA, 3nm for the flexible and 30 nm

for the rigid part are not realistic at all. A random sequence has a typical flexibility of 50 nm and, to my knowledge, 30 nm is already a quite flexible DNA sequence. For the chromatin, the values of 10nm (ie 5 nucleosomes, since they consider a 1000kbp bead size) and 100nm (10kbp) seem to be more relevant. But authors do not justify their choice (from experiments? modeling of the chromatin fiber?). The authors should show how their findings depend and the choice of eu vs heterochromatin values (the difference between eu- and hetero- values, as well as the size and distribution of the domains surely control the strength of the phase separation).

8) Again, I don't understand the model of integration: which (bending?) energy cost is used to model the integration step? For the DNA, I can guess that it is the same as in the first section (but again, I would like to know what bending constraint is used, see point 1)). But for the chromatin model? Why local integration of a viral DNA should also depend on the local flexibility of the chromatin fiber? If it is the case, authors should again detail the model.

To be consistent with their “small scale” model, the integration would be rather promoted in a nucleosome dense chromatin, thus heterochromatin. However, if I understand well, their model rather promotes integration in the flexible parts that they associate with the euchromatin. There is thus a contradiction if we consider the small scale model they proposed in the first section. Actually, the observed preferential targeting to euchromatin might be due to the presence of cofactor that indeed promote integration in active parts (nothing to do with DNA or chromatin flexibility); then final insertion might be driven by elasticity-driven indirect read-out. In addition, a too dense array of nucleosomes might be not favorable due to steric hindrance. A more open chromatin (like euchromatin) would be then more favorable. All this argue in favor of developing a model of integration at the scale of the chromatin fiber that goes beyond the DNA bending constraint !

Authors should also explain why they consider that euchromatin is more flexible than heterochromatin. But again I don't see why flexibility is the relevant parameter. I would rather have chosen local chromatin fiber compaction and then introducing a model that relate integration to this compaction level.

9) I don't really get the objective of this section and in particular whether or not authors want to address the influence of 3D organization vs local flexibility in the final integration landscape ? I may have missed something: due to the di-block nature of the chain, with the alternation of highly flexible and rigid subchains, there is a phase separation that leads to an enhancement of integration in the flexible domains because of a higher accessibility to these domains that form “petals” surrounding the globular phase made by rigid domains. Is that right ? The authors should thus show how this phase separation enhances integration in the flexible part as compared to the unfolded case (not as compared to the uniform “expected” integration case, Fig 2E).

10) “In the context of naked DNA, this is in agreement with...(hence different flexibility)[9]”: Preferential integration in some specific more flexible DNA sequences is a local effect and certainly not due to the spatial folding of the whole chain. As already pointed out, the DNA is an heteropolymer with flexibility variations that are not as large as they introduced in their model. Such chain cannot experiment a phase separation, so selectivity is only question of the primary sequence. I would remove this sentence.

11) “In Fig2C we also (?) a kymograph... “: The authors notice that there are some “hierarchical integration events”, showing one example in Fig2 C but they don't provide any quantification and any explanation of these events. I think that it has to done.

12) “We further analyse...”: Is there any experimental evidence of such scaling behavior ? What do we really learn from this law ? There is no discussion concerning the biological relevance of such prediction.

“Non-Equilibrium Integration Process Yields a Uniform Distribution of Integration Sites”

13) It seems to me that it is a straightforward result that does not need any simulation. I would remove this part. Furthermore their conclusion (“this may lead the intasome complex to target heterochromatin regions...”) strictly depends on their model of final integration; it doesn’t hold in a model where the preferential targeting to the euchromatin is due to some preferential interaction of the intasome with transcriptional activator, or active marks (or in a model where the open structure of euchromatin favor integrase binding activity). In such models even if the final integration of the viral DNA is a non-equilibrium process, integration will still preferentially occur in euchromatin (but in euchromatin, the pattern will become indeed uniform).

“Large-Scale 3D Chromosome Folding Enhances Euchromatic Integration”

14) As in the previous section, in this section the author investigate the role of phase separation between eu- and heterochromatin in the process of integration. In that case, phase separation results only from self-attraction between heterochromatin beads (since they assume here a same flexibility). Since integration is now equi-probable in eu vs heterochromatin, they can test the influence of the phase separation in the integration selectivity. This is conceptually the same story as before but with the demonstration that a phase separation is sufficient to drive integration toward euchromatin, whatever the mechanisms of integration. As for the flexibility-driven phase separation, authors consider parameter that lead to a rather strong phase separation: how they justify the choice of $3kT$ for heterochromatin self-attraction ? It seems to me that heterochromatin is in a fully collapsed state (equilibrium globule). Is it relevant ? More generally, does the obtained spatial phase separation correspond to the in vivo situations ? Is there any Hi-C/microscopy experiment to compare with ?

In Fig E,F they should add the distribution of eu- and heterochromatin domains. The agreement with experiment is indeed good but I would prefer a comparison with their model of flexibility-driven integration in the unfolded case when considering different flexibilities (eu vs hetero) as in the previous section (in order to see whether or not a model with only local preferential integration in euchromatin is less efficient).

“Observed Distribution of HIV Hot-Spots in T-Cells is Recapitulated by a Refined Model”

15) It might be relevant, in their “single-cell” refinement, to consider in the model the distribution of nuclear pores at the periphery since they constitute the entry gate inside the nucleus. Such pore distribution would probably fix the boundaries between eu and heterochromatin shells (exactly as in the inset of Fig 5E).

Response to Reviewers' comments:

To address all of the Reviewers' reservations, we have now performed a substantial amount of extra work and rewritten the manuscript considerably. First, we have added a new Figure (Fig. 2 in the revised version) to study HIV integration on polynucleosome chromatin fibres (as advised by Reviewer #3). This provides a natural link between the mononucleosome results (Fig. 1) and the full-chromosome simulations (Fig. 3). Second, we have cut the whole section of HIV integration in diblock chromatin fibres, as after reading Reviewer #3' comments, we realised it might have been confusing for a reader. Third, we have addressed all other remaining comments of Reviewer #1 and Reviewer #3, in particular discussing in detail the potential impact of other integration factors not included in our modelling, and of supercoiling. All main changes in our revised version (not including minor edits) are highlighted in blue.

We are convinced that the revised version is now significantly improved, and that the new work has both strengthened our conclusions and demonstrated the validity of our approach, which leads to results matching a large number of experimental observations on HIV integration.

We would like to thank the Reviewers for their insightful comments on our work which prompted us to make these revisions. We would hope that they will now be willing to comment on our revision, and trust that they will see all their comments have been constructively addressed. Below is a point-by-point reply to all of the Reviewers' comments.

With best wishes,

D. Michieletto, M. Lusic, D. Marenduzzo, E. Orlandini

Reply to Reviewer #1

COMMENT:

In this manuscript, the authors have studied how HIV-1 integration can be regulated by physical parameters of the nucleus, at different scales of its organization. Using modeling tools, they test how specific physical properties of the cellular DNA and chromatin (such as its elasticity, flexibility or the attraction between distinct chromatin regions) affect the selectivity of integration. They also use a reaction-diffusion model to study the localization of the provirus in the nucleus. Their modeling approaches allow the authors to reproduce physiological specificities of HIV integration sites such as their enrichment in DNA regions covered by nucleosomes, in flexible genomic regions, in euchromatin versus heterochromatin, close to integration sites mapped in infected cells and in euchromatin regions of the nucleus.

This is an original and interesting study, with both fundamental and therapeutic impacts.

However, I have three major reserves about this manuscript.

RESPONSE:

We thank the referee for the encouraging and positive comment about the impact of our work. Indeed, to the best of our knowledge, this is the first polymer model for HIV integration. We now provide a constructive point-by-point rebuttal to all her/his reservations.

COMMENT:

- First, most of the predictions have been made without taking into account (and even without citing corresponding studies) several cellular and viral parameters (like LEDGF/p75, Cpsf6 and specific nuclear pore proteins) that have been

demonstrated to be important for the selectivity of HIV integration. Some of these proteins can modulate this process at the nucleosome level and other ones at the level of nuclear organization. The authors could at least have discussed the relevance of their results with regards of several virology studies on this process.

RESPONSE:

We thank the referee for raising this point. Indeed, there are many proteins and enzymes involved in retroviral integration and nuclear organisation in general. Yet, experimental evidence, such as that reported in Pruss et al, PNAS 1994, shows that it is possible to achieve HIV integration and its bias towards flexible/nucleosomal DNA in vitro without the need of any enzyme other than integrase. Other works, such as Ciuffi et al, Nature Medicine 2005, show that in cells where LEDGF/p75 is knocked-down, the profiles of HIV integration are still significantly biased towards euchromatin. All this points to the fact that there may be an intrinsic bias in HIV integration site selection that cannot be explained by invoking cellular cofactors.

In our work we show that thermodynamics and accessibility can fill this gap and recapitulate some of the current unexplained evidence. We now highlight this important aspect of our work in the revised text.

At the same time we fully agree with the reviewer that these factors are important. To constructively address the referee's comment in the revised text we therefore now discuss the role of these factors in some detail in the Discussion section.

COMMENT:

- Second, in the initial model of integration, the viral DNA is mimicked by a circular DNA but no precision is given about its length, its covering by proteins and the presence of the intasome which connects the LTR ends. In structures of this complex (solved by crystal or cryo-em approaches), the two LTR ends form an acute angle and the target DNA is also highly bent. These parameters should be taken into account or at least mentioned to explain how they could affect the selected modeling parameters.

RESPONSE:

As now clarified in the revised text, because the integration of the viral DNA (vDNA) into the host is a local process that occurs on essentially 1 base-pair, the total length of the viral DNA is not important (in general we consider viral DNA as loops made of 40 beads or 320 bp, for computational efficiency).

We agree with the reviewer that the vDNA geometry within the intasome may play an important role and should be considered in the modelling. Hence, to address the referee's comment we have now performed new simulations where one site along the vDNA is kinked thus mimicking the presence of the intasome joining the LTR ends to form an acute angle. As we show in the new Fig. 1, this refinement does not affect our results; yet, as we show in the SI, our new simulations show that in this refined model the integration along the vDNA is strongly preferred at the kink (where this has to occur experimentally). This result has, we believe, added validity to our modelling. We thank the referee for prompting us to include this aspect in our model.

COMMENT:

- Third, if the selected physical parameters are sufficient to reproduce major specificities of HIV integration, why are these specificities different between HIV and other retroviruses?

RESPONSE:

This is a valid and interesting point. Many families of retroviruses display a preference for active chromatin and flexible/nucleosomal DNA substrates and these are captured by our generic model. Indeed, in our model, all retroviral integrases that work in quasi-equilibrium (as more precisely explained in our revised text) should display similar integration patterns. Yet, at our level of coarse-graining we cannot distinguish between integration within enhancers,

promoters, body of the genes or transcriptional start sites. These fine details may be attributed to the action of other cofactors, and we now discuss them in the revised text (in the Discussion section).

One important exception to the cases discussed above is provided by the families of alpha- and beta-retroviruses (such as the mouse mammary tumour virus, MMTV). These retroviruses display a markedly unique integrase enzyme composed by 8 subunits which can accommodate unbent tDNA. Further, they cannot traverse the nuclear envelope and thus target cells during mitosis, when the large-scale chromatin organisation is very different from interphase. We thus expect our simulations not to represent these families of retroviruses, yet some of these peculiar features may be accounted for in future works. We thank the referee for this stimulating question and we now provide a detailed discussion of these aspects in our revised work.

COMMENT:

For these reasons, but also because the authors have used very precise physical models, I propose to submit this manuscript to a more specialized journal. I also suggest to insert a more detailed discussion on the biological relevance of the selected modeling parameters and the pertinence of the obtained results with regards of previous studies published in the field of HIV integration.

RESPONSE:

The referee's conclusion seems at odds with his premise that "this is an original and interesting study, with both fundamental and therapeutic impacts". Also, we strongly feel that our work should not be dismissed because it uses precise physical models. Instead, this is a strong point of our work, being the first to lay down a theoretical biophysical model for HIV integration. As we have now constructively addressed all of the referee's comments, we trust that she/he will find our revision now suitable for publication in Nature Communications.

Reply to Reviewer #2

COMMENT:

I think this manuscript discusses a very nice and novel idea on the mechanisms of viral DNA integration in the human genome. The physics model here introduced is very interesting and well explained. Its implementation is technically demanding, but well tackled by the authors. The results are very interesting, especially considering the simplicity of the considered model. They are in good overall agreement with available experimental data. As a minor comment, I would advice the authors to represent even better the framework of their study and the corresponding literature. In brief, I definitely support this manuscript for publication.

RESPONSE:

We are very grateful to the Reviewer for her/his strong endorsement of our work. We have now significantly expanded our discussion of the model and also expanded the references to relevant work in HIV virology.

Reply to Reviewer #3

COMMENT:

"In their manuscript, entitled "Multi-scale Model Reveals Physical Principles of HIV Integration in Human Nuclei", Michieletto and colleagues propose a theoretical modeling of HIV integration at different scales, from DNA and nucleosomes to the nuclear 3D organization. This article constitutes to my knowledge one of the first attempt to investigate HIV integration with simple theoretical framework. But as mentioned in my comments listed below, I am not fully convinced by their models (too simplistic) or by the relevance of their

investigation, except for the very large scale. The models are not sufficiently well explained and their parametrization not enough justified. Actually there is almost no cross-talk between the scales: it is rather a "multiple-scale" approach than "multi-scale". In addition there is no model of integration at the scale of the nucleosomal array and chromatin fiber which is, to my opinion, the most relevant scale to investigate. There are many points that have to be clarified. This work is actually too ambitious to provide satisfactory results at all scales; I would suggest to investigate the integration process scale by scale (in independent papers) but more deeply and exhaustively. For these reasons and the concerns listed below I cannot recommend this article for publication in Nature Communication."

RESPONSE:

We thank the referee for her/his acknowledgement that our work is among the first attempts to model HIV integration with a theoretical framework. There has indeed been - to the best of our knowledge - no prior polymer models for HIV integration: this is a key strength of our work, and the major reason why we strongly feel that Nature Communications, with its broad readership which includes physicists and biologists is the ideal avenue to communicate our results.

We disagree that our framework should be dismissed as "too simplistic". Our aim here is to find the key ingredients that are necessary to rationalise the observed site-selection of HIV integration. Our model can capture the essential physics that is involved in the process and this allows us to obtain integration patterns that are in quantitative agreement with experiments at several scales. Therefore the apparent simplicity of the model, and - importantly - the absence of fitting, is a strength, not a weakness, of our work.

At the same time, we are very thankful for the insights offered by the referee; they stimulated us to critically review some of our models and to perform new calculations, or to clarify the text when this created some misunderstandings. Specifically, we have included a more substantial section on the model description, and, more importantly, a new Figure and section on integration on polynucleosomal array. Following the feedback from the reviewer, we have also decided to remove the potentially confusing section on di-block chromatin fibres, so that several of the referee's objection no longer apply (see points 6-12 below).

As a consequence of this reformatting, we believe that our revised paper is significantly improved, and the focus has narrowed as advised by this Reviewer and the Editor. Below, we provide a constructive point-by-point rebuttal to all the reviewer's comments and questions.

"DNA Elasticity Biases HIV Integration"

COMMENT:

"1) The model of DNA integration is not clearly explained enough. In particular, the authors should describe more precisely the elastic constraint(s) (bending -and twisting ?-) imposed to the viral and host DNA during the insertion process. I agree that such constraints coming from the local DNA conformation (the viral/host DNAs or/and the final host+viral DNA ?) imposed in the intasome by the integrase complex may provide an indirect read-out mechanism for intasome targeting and viral DNA insertion. But, in their model and numerical simulation, what is the exact bending constraint and thus the value of the bending cost that controls final insertion ? Is it the local bending at the two junctions of the final ligated DNA ? "

RESPONSE:

We now discuss our integration model in more detail in a dedicated section of our revised text. Briefly, target and viral DNA (tDNA and vDNA) are treated as semiflexible polymers with persistence length=50 nm, as appropriate for naked DNA. In our model, the integration process is condensed into one key step which is a stochastic recombination of 3D-proximal tDNA and vDNA strands. In this

step, the energy difference between new (host+viral) conformation and old (host and viral) conformations is calculated and used to assign a probability with which to accept the integration move. A major contribution to the energy difference is - as argued by the reviewer - the local bending of the DNA, before and after ligation. The energy difference is also determined by stretching of the bonds between beads (equivalent to longitudinal deformations along the DNA backbone), and by steric interactions (equivalent to excluded volume effects). Importantly, while we agree that our model only gives a simplified representation of HIV integration, this approach fully includes stochasticity (and thermodynamics), which is necessary to probe the free energy landscape along the tDNA. This is particularly important since IN needs to carry out integration without ATP. We believe that this is now all clarified in the revised text.

COMMENT:

"Actually, I thought (see paper Michel F, Crucifix C, Granger F, Eiler S, Mouscadet JF, Korolev S, et al. Structural basis for HIV-1 DNA integration in the human genome, role of the LEDGF/P75 cofactor. The EMBO journal. 2009;28(7):980-91.) that the targeting of the intasome at the host DNA was rather driven by the propensity of the host DNA to be bent (providing a conformation favorable for final ligation with viral DNA, conformation imposed by the integrase complexe). However it seems to me that the model proposed by the authors does not account for this constraint at this stage of integration but rather for the bending constraint at the ligation stage. I am not sure that it is equivalent. Authors should clarify this point by illustrating, for example, a typical insertion process (the drawing in Fig 1 E,F are not very clear; for example, what are the two depicted integration products, how are they selected etc...?) and showing which step is controlled by a bending energy activation barrier and of which amplitude? The exact geometry (bending angle) of the constrained DNA should be more clearly illustrated. Actually, there might exist different logics of insertion by such indirect readout process depending on the integrases+viral DNA complexes architecture. It could be interesting to discuss that point and by this way to justify their specific choice for the imposed constraint."

RESPONSE:

We believe that the revised text now clarifies this point and we provide new clearer illustrations of the process (see also point above, and new Fig. 1). At the same time, one issue with the reviewer's argument here is that it is mainly based on structural considerations, and disregards thermodynamical ones (due to configurational fluctuations), which are instead crucial at least in our model. The key thing is that our approach is based on the *difference* of energy in a configuration following and before the integration. Therefore, the energy barrier in the typical insertion process in Figure 1B is simply the difference in energy post and pre-insertion (the detailed sequence of intermediate structures between the two is not important, only the barrier value is). Thermodynamically, there is a distribution of possible energy barriers in our model (as the polymer configurations fluctuate). The integration profiles we find are due to the difference in the typical energy barrier in different locations along the tDNA. Therefore, for instance, the model naturally reproduces also the preference for insertion into kinked DNA as mentioned by the reviewer. This is a simple extension of the new results which we now show in the SI, where we explicitly study the case of integration along a naked DNA segment split into one with typical rigidity ($l_p=50$ nm) and one with a flexible ($l_p=30$ nm). We agree that it is important to relate to the literature cited by the reviewer, hence we have expanded its discussion, in relation to the papers she/he has highlighted.

COMMENT:

"2) The authors have not considered twist constraints: is there any argument for that? Depending on the geometry of the viral DNA in the intasome, a twist deformation might also be applied to the host DNA in order to have the proper chemical phasing (in addition to the proper duplex orientation). Twist (and

supercoiling) might be also relevant to understand nucleosome selectivity."

RESPONSE:

The referee raises a good point here, as twist and supercoiling may in principle play a role in our problem.

Twist constraints are notoriously difficult to implement on a coarse-grained model for DNA and here we chose to neglect them for simplicity. The quantitative agreement with experiments suggest that perhaps twisting plays a secondary role with respect to bending. Regarding supercoiling, we highlight here that a key feature of supercoiled DNA loops is that they develop writhing, which entails bending deformations in 3D. Such deformations are focused on the tip of the plectonemes which appear in supercoiled molecules. Writhing will broadly favour integration (as for the case of the histone described in Fig. 1).

Thus, our results lead to two qualitative predictions for supercoiled DNA:

(i) integration should be globally more likely within supercoiled DNA with respect to torsionally relaxed naked DNA, in view of the enhanced writhing (and bending) in the former;

(ii) integration within a supercoiled DNA should be especially likely locally in the highly bent plectoneme tips.

Experiments with naked relaxed and supercoiled DNA have confirmed the prediction in (i), whereas we are not aware of any experiment that tested (ii), which we therefore present as a downstream prediction of our model, testable in the future. We thank the referee for directing us towards this interesting aspect of our work and we have now added a section to discuss this aspect in the revised manuscript.

COMMENT:

"3) As mentioned in the section "A Di-Block..."..., DNA is an heteropolymer whose elastical properties depend on the underlying base sequence. There are sequences that are more flexible or instrinsically bent and that, consequently, may also favor viral DNA integration. So there might be also a bias toward specific sequences even for naked DNA by the same indirect read out process. Authors discuss that point in the next section but I would recommend to discuss this sequence effect here."

RESPONSE:

Following this and some subsequent comments of this reviewer, we have now critically reviewed the whole section previously entitled "A di-block ...", and decided to drop it entirely. This has also helped narrowing the focus of our work which is one of the general pieces of advice this reviewer has given us. As a result Figure 1 is indeed the right place to describe effect of DNA flexibility on integration, as suggested by the reviewer. Therefore, in the new version of Figure 1 we now include new simulations and previous experimental results which directly address sequence-dependent flexibility in presence and absence of nucleosomes. In addition, in the SI (Fig. S1) we report new simulations where we explicitly study the integration statistics along naked DNA with heterogeneous flexibility and show that, indeed, our model captures the known HIV bias for flexible DNA.

COMMENT:

"4) Concerning the preferential integration in the mononucleosome vs naked DNA, indeed the curved configuration imposed by the wrapping around the octamer may favor the final bent configuration inside the intasome (what about twist ?) but is there a good reason to neglect the interaction between the intasome complexe and the nucleosome ? There might be a steric hindrance that would rather inhibit integration but reversely, there might be a specific positive interaction between integrase (or associated cofactors) and the histones , that would promote the targeting at nucleosome and final integration. In their model, preferential integration is at the dyad position: is there any experimental evidence for that ? DNA breathing at the entry-exit might also favor integration due to enhanced accessibility. If there are (in vitro) datas of integration (mono-nucleosome vs naked DNA) authors should show to what extend their simple model indeed reproduce them."

RESPONSE:

The referee is correct in thinking that this finding is (at least partially) counterintuitive. The nucleosome structure should hinder accessibility and DNA breathing at entry/exit points should enhance accessibility of the intasome. Yet, long standing evidence (Pruss et al, "Human immunodeficiency virus integrase directs integration to sites of severe DNA distortion within the nucleosome core", PNAS (1994) & Pruss et al, "The Influence of DNA and Nucleosome Structure on Integration Events Directed by HIV Integrase", J Biol Chem (1994)) show that this is not the case. HIV integration performed by IN displays a preference for histone-bound DNA. Of course there could be specific interactions between the nucleosome and the IN structure that explain this observation. Yet our model is simpler in that does not need to invoke other factors or specific intasome-histone interactions to explain the current evidence. Also, a model with specific positive interactions cannot explain the different integration patterns in DNA substrates with different flexibilities. On the contrary our model can. We have now added a discussion on other possible biochemical factors favouring integration in nucleosomal DNA.

Regarding dyad preference: we recall that our model captures both partial nucleosome unwrapping and steric accessibility. The observed preference for the dyad, in our interpretation, is due to the fact that such unwrapping decreases the local bending, which is the main factor affecting integration (accessibility in this particular case appears to play a minor role). By symmetry, the inner central segment is the one that is most likely to be histone-bound at any time and it is thus the one that is most bent - hence most targeted. Our model neglects other relevant details such as minor-major grooves and charge distribution on the histone octamer; for this reason we do not expect to recapitulate the distribution of integrations at the level of few base-pairs. We now discuss this in the text.

Regarding the comparison with experiments: we now provide a detailed quantitative comparison with the data on integration on nucleosomal DNA versus naked DNA in silico by Pruss et al. ("Human immunodeficiency virus integrase directs integration to sites of severe DNA distortion within the nucleosome core", PNAS (1994)). As shown in the new Fig.1D, our simulations are in quantitative agreement with their results.

COMMENT:

"5) In vivo, the relevant substrate for integration is the nucleosomal array. Even when considering the possibility of integrating in the naked linker DNAs, the presence of neighbouring nucleosomes may lead to a very different situation than integration in a completely naked DNA. Nucleosomal array and the chromatin fiber is, to my point of view, the very relevant scale to investigate: it is the scale of integrase binding and processing. Furthermore, large scale studies would require to have a model of integration at this scale that could be eventually coarse-grained. This is currently not the case (see next section)."

RESPONSE:

We thank the reviewer for this great suggestion. We have fully taken this point on board, and we have now added a new Section and Figure (Fig. 2) where we consider the case of a nucleosomal array. We agree that this intermediate step, which was previously omitted, is very important, and we now consider this instead of the differential flexibility fibre which was previously discussed in the original submission. Integration on a polynucleosomal array also leads to a new result, which is again due to the combination between the two main drivers for integration: the targeting of elastically bent DNA and the geometric issue of accessibility. Indeed, there is a weakening of preference for nucleosomal (over linker) DNA, which depends on nucleosomal density, similarly to what was expected by the reviewer. We thank again the reviewer for directing us towards this previously unexplored aspect of our model.

"A Di-Block Polymer Model Predicts Preferential Integration into Flexible Genomic Regions"

COMMENT:

"6) In this section, I actually don't understand why the authors consider both the DNA and the chromatin systems ? I think that all could be done with a chromatin model. The main reason is that there is no large portion of naked DNA in vivo, so it might be not so relevant to study integration within naked DNA at large scales. The other reason (see point 7)) is that I do not believe that DNA can experiment phase separation due to flexibility inhomogeneity."

RESPONSE:

We have now removed this section from the main text as this and subsequent observations made us realise that it may create confusion in the reader. Essentially, before we proposed that the same model could be considered for DNA-only (in vitro) or chromatin-only (in vivo) - no mixed system was considered.

We now drop the di-block polymer model, and address flexibility effects on integration within DNA in vitro in Figure 1 and in the SI Fig. S1. In the main text we show that integration is more likely in nucleosomal DNA, but that this effect is weakened when flexible DNA is considered (in agreement with experiments). In the SI, we explicitly study the case of naked DNA with heterogeneous flexibility.

With this reformatting of our work there should be no more room for misunderstandings - at the same time, we highlight that our results did not depend (and we never claimed they did) on phase separation driven by flexibility inhomogeneity. Although this is a known effect (see Cook & Marenduzzo, "Entropic organization of interphase chromosomes" J Cell Biol (2009)), in our case the different energy barrier against local deformations is the main driver for HIV site selection.

COMMENT:

"7) Concerning the di-block copolymer model: for DNA, 3nm for the flexible and 30 nm for the rigid part are not realistic at all. A random sequence has a typical flexibility of 50 nm and, to my knowledge, 30 nm is already a quite flexible DNA sequence. For the chromatin, the values of 10nm (ie 5 nucleosomes, since they consider a 1000kbp bead size) and 100nm (10kbp) seem to be more relevant. But authors do not justify their choice (from experiments ? modeling of the chromatin fiber ?). The authors should show how their findings depend and the choice of eu vs heterochromatin values (the difference between eu- and hetero- values, as well as the size and distribution of the domains surely control the strength of the phase separation)."

RESPONSE:

Those values of flexibility were chosen indeed as relevant for chromatin, as eu and heterochromatin are normally considered to be on opposite ends of the flexibility spectrum (Cook & Marenduzzo, "Entropic organization of interphase chromosomes" J Cell Biol (2009), and references therein). The results depend on the ratio between persistence lengths rather than on the absolute numbers. However we have now eliminated this section to make room for the polynucleosome chain simulation, so these issues are no longer relevant. In the revised SI, we now consider naked DNA with correct persistence lengths, i.e. $l_p=50$ nm for typical DNA substrate and $l_p=30$ nm for a flexible one and show that our results still hold.

COMMENT:

"8) Again, I don't understand the model of integration: which (bending ?) energy cost is used to model the integration step ? For the DNA, I can guess that it is the same as in the first section (but again, I would like to know what bending constraint is used, see point 1)). But for the chromatin model ? Why local integration of a viral DNA should also depend on the local flexibility of the chromatin fiber ? If it is the case, authors should again detail the model."

RESPONSE:

We refer to the points above (and the revised text) for details on the integration model.

As previously mentioned, we now critically reviewed our diblock model and agree

with the referee that our interpretation of integration along a di-block chromatin fibre with heterogeneous flexibility is a simplified description of chromatin and might be potentially confusing for a reader, hence this di-block section has been eliminated (see above).

COMMENT:

"To be consistent with their "small scale" model, the integration would be rather promoted in a nucleosome dense chromatin, thus heterochromatin. However, if I understand well, their model rather promotes integration in the flexible parts that they associate with the euchromatin. There is thus a contradiction if we consider the small scale model they proposed in the first section. Actually, the observed preferential targeting to euchromatin might be due to the presence of cofactor that indeed promote integration in active parts (nothing to do with DNA or chromatin flexibility); then final insertion might be driven by elasticity-driven indirect read-out. In addition, a too dense array of nucleosomes might be not favorable due to steric hindrance. A more open chromatin (like euchromatin) would be then more favorable. All this argue in favor of developing a model of integration at the scale of the chromatin fiber that goes beyond the DNA bending constraint!

Authors should also explain why they consider that euchromatin is more flexible than heterochromatin. But again I don't see why flexibility is the relevant parameter. I would rather have chosen local chromatin fiber compaction and then introducing a model that relate integration to this compaction level."

RESPONSE:

The new simulations on polynucleosomes confirm the referee's intuition, as previously described (see new Fig. 2). In particular, the total number of integration events decreases in condensed nucleosomal arrays and integration appears to target linker DNA as well as than nucleosomal one. The balance between nucleosomal and linker DNA integration depends on fibre density and is discussed in the text in detail. Additionally, having dropped the section on di-block copolymers the flexibility at the level of chromatin fibre is no longer pertinent - we agree that its relevance would merit a deeper discussion. We still believe the dependence on local chromatin flexibility to be an interesting feature to model, and we will come back to this in the future, but it is now outside the scope of the current work.

COMMENT:

"9) I don't really get the objective of this section and in particular whether or not authors want to address the influence of 3D organization vs local flexibility in the final integration landscape ? I may have missed something: due to the di-block nature of the chain, with the alternation of highly flexible and rigid subchains, there is a phase separation that leads to an enhancement of integration in the flexible domains because of a higher accessibility to these domains that form "petals" surrounding the globular phase made by rigid domains. Is that right ? The authors should thus show how this phase separation enhances integration in the flexible part as compared to the unfolded case (not as compared to the uniform "expected" integration case, Fig 2E)."

RESPONSE:

This issue - which is no longer relevant as we have now dropped the di-block section - originated from a misunderstanding. The daisy-like structure and phase separation of hetero- and eu-chromatin is observed within the model that includes self-attractive interactions and uniform flexibility, *not* in the previous di-block model which considered no attraction and differential flexibility. The effect of flexibility on integration were due to local changes in the bending energy and not to phase separation. We trust that this is no longer an issue with the new format of the manuscript.

COMMENT:

"10) "In the context of naked DNA, this is in agreement with...(hence different flexibility)[9]": Preferential integration in some specific more flexible DNA sequences is a local effect and certainly not due to the spatial folding of the whole chain. As already pointed out, the DNA is an heteropolymer with

flexibility variations that are not as large as they introduced in their model. Such chain cannot experiment a phase separation, so selectivity is only question of the primary sequence. I would remove this sentence."

RESPONSE:

See points above. DNA flexibility is now included in Figure 1 and SI Fig. S1.

COMMENT:

"11) "In Fig2C we also (?) a kymograph... ": The authors notice that there are some "hierarchical integration events", showing one example in Fig2 C but they don't provide any quantification and any explanation of these events. I think that it has to done."

RESPONSE:

These hierarchical integration events, which are potentially interesting, are no longer discussed in the revised version. This effect may be due to the relative fraction of viral genetic material compared to the total target DNA, which is here much larger than in real situations for computational efficiency. With the aim of narrowing the focus of this work, we now decide to avoid mentioning these hierarchical integration events.

COMMENT:

12) "We further analyse...": Is there any experimental evidence of such scaling behavior ? What do we really learn from this law ? There is no discussion concerning the biological relevance of such prediction.

RESPONSE:

This scaling was discussed in relation with the diblock simulations, now eliminated. We agree that the study of the scaling while interesting was a secondary result, and again in the interest of focussing on the main new message now we no longer discuss the dependence on volume fraction in the revised version. Nonetheless, we note that the dependence of integration rate on total chromatin density may be relevant for quantifying infection statistics in cell lines with nuclei of various dimensions.

"Non-Equilibrium Integration Process Yields a Uniform Distribution of Integration Sites"

COMMENT:

"13) It seems to me that it is a straightforward result that does not need any simulation. I would remove this part. Furthermore their conclusion ("this may lead the intasome complex to target heterochromatics regions...") strictly depends on their model of final integration; it doesn't hold in a model where the preferential targeting to the euchromatin is due to some preferential interaction of the intasome with transcriptional activator, or active marks (or in a model where the open structure of euchromatin favor integrase binding activity). In such models even if the final integration of the viral DNA is a non-equilibrium process, integration will still preferentially occur in euchromatin (but in euchromatin, the pattern will become indeed uniform)."

RESPONSE:

We now provide an example of a purely non-equilibrium integration mechanism in comparison with quasi-equilibrium one in the SI (Fig. S1). This is done on segments of naked DNA with heterogeneous flexibility. As expected, purely non-equilibrium integration does not display the bias towards flexible segments. To constructively address the referee's comment, we now limit the discussion of this effect to the SI. Yet, we would like to stress once more that our model can explain a range of findings with only one hypothesis. On the other hand invoking the action of cofactors or preferential interactions of the intasome would require one cofactors/interaction for each finding, including flexible/curved DNA. We believe that our model is simpler and more generic and yet it can explain much of the current evidence.

We also mention that although considering a purely non-equilibrium strategy may lead to trivial results, it helps raising an interesting implication of our

work, i.e. that if HIV integration works in quasi-equilibrium then its integration statistics needs to be affected by thermodynamics.

"Large-Scale 3D Chromosome Folding Enhances Euchromatic Integration"

COMMENT:

"14) As in the previous section, in this section the author investigate the role of phase separation between eu-and heterochromatin in the process of integration. In that case, phase separation results only from self-attraction between heterochromatin beads (since they assume here a same flexibility). Since integration is now equiprobable in eu vs heterochromatin, they can test the influence of the phase separation in the integration selectivity. This is conceptually the same story as before but with the demonstration that a phase separation is sufficient to drive integration toward euchromatin, whatever the mechanisms of integration. As for the flexibility-driven phase separation, authors consider parameter that lead to a rather strong phase separation: how they justify the choice of $3kT$ for heterochromatin self-attraction? It seems to me that heterochromatin is in a fully collapsed state (equilibrium globule). Is it relevant? More generally, does the obtained spatial phase separation correspond to the in vivo situations? Is there any Hi-C/microscopy experiment to compare with?"

RESPONSE:

We recall that, since we had not looked at phase separated systems in the previous stiff/flexible di-block section (now eliminated), the results we report here are not "the same story". As the reviewer points out, we consider here equiprobable integration distribution along the polymer (as the flexibility is uniform) so the only driver for site-selection is large-scale folding (connected to geometric accessibility, or steric hindrance between folded chromatin and viral DNA, assumed here to be chromatinised in agreement with experimental studies, D. M. Knipe et al., "Snapshots: Chromatin control of viral infection", *Virology* (2012)). The self-association parameter for heterochromatin-heterochromatin interactions is here chosen based on our previous works (see Brackley et al, *PNAS* (2013); Brackley et al, *NAR* (2016); Michieletto et al, *PRX* (2016)) where we show quantitative agreement between simulated and experimental HiC. Since this comparison is not the key point of this paper, we avoid discussing this aspect in detail and refer the reader to our previous works.

COMMENT:

"In Fig E,F they should add the distribution of eu- and heterochromatin domains. The agreement with experiment is indeed good but I would prefer a comparison with their model of flexibility-driven integration in the unfolded case when considering different flexibilities (eu vs hetero) as in the previous section (in order to see whether or not a model with only local preferential integration in euchromatin is less efficient)."

RESPONSE:

We have now added the location of eu- and heterochromatin domains in Fig. 3E, F.

We have now dropped our di-block polymer model with heterogeneous flexibility so this comparison is no longer relevant and would be confusing. Here the main driver is accessibility. For completeness, we report that the accessibility-induced bias towards euchromatin integration is larger than the one which would be found due to the flexibility effect in a di-block stiff/flexible chromatin model such as the one previously considered.

"Observed Distribution of HIV Hot-Spots in T-Cells is Recapitulated by a Refined Model"

COMMENT:

"15) It might be relevant, in their "single-cell" refinement, to consider in the model the distribution of nuclear pores at the periphery since they constitute the entry gate inside the nucleus. Such pore distribution would probably fix the boundaries between eu and heterochromatin shells (exactly as in the inset of Fig

5E)."

RESPONSE:

Our model for the whole nucleus could be adapted to single cell only if the precise position of eu- and hetero- chromatin was exactly known. In this case, the position of nuclear pores would also provide an essential information. Instead, our model can be more easily applied to generic cells within a population where there is natural variation in eu-,hetero-chromatin and nuclear pores location. In this case there is no need to know the precise location of these elements in all the cells, but only their average position. This is also the most relevant case to compare with experiments, as RIGs are measured on a population-wide scale. In the future, we aim to improve this model so that it can be used to map hotspots in single cells starting from minimal information. We expect that the location of nuclear pores will be, as the referee suggests, a crucial piece of information in that case but here is not needed to explain the experimental results. We have commented on this in the revised text.

Since we have constructively addressed all of the referee's comments we hope that now she/he will reconsider our work for publication in Nature Communications.

Reviewer #2 (Remarks to the Author):

I think that the authors have replied in full details to the issues raised by referee #2 and #3. They have taken into account, in particular, all the useful comments and requests by referee #3. I still think the paper should be published.

Reviewer #4 (Remarks to the Author):

Transparency statement: my name is Ruggero Cortini, I am a theoretical biophysicist. I can assess the part of the manuscript that concerns the physical modelling and its biophysical implications. I have limited understanding of HIV biology. I was asked to be a referee at the first revision stage of the manuscript.

The manuscript "Physical Principles of HIV Integration in the Human Genome" by Michieletto et al illustrates a study based on theoretical physics of how HIV integrates in the human genome. There are three main parts of the paper: in the first, the authors propose a quasi-equilibrium stochastic model based on computer simulations for HIV integration, and analyse the properties of physical integration of a plasmid in a model of a chromatin fibre; in the second, they go up a length scale and analyse within the same model how integration is biased towards different regions of the genome modelled as euchromatin or heterochromatin; finally, they propose a reaction-diffusion analytical framework to understand experimental patterns of HIV integration as a function of the distance of the genome from the nuclear membrane.

The main findings can be recapitulated as follows. First, the Authors are able to explain experimentally observed biases of HIV integration towards chromatin, flexible DNA regions, and curved DNA segments, because of straightforward energetic considerations. In their model in fact the elastic energy of the juxtaposed segments of the modelled polymers (vDNA for viral DNA, tDNA for target/host DNA) plays a pivotal role in determining the probability for an integration event. The second finding is that HIV prefers to integrate in euchromatic regions of the genome, because those regions are found in external "petal-like" structures which are readily accessible to the viral DNA. Finally, the Authors are able to reproduce accurately the patterns observed in a recent experimental work on the probability of HIV integration as a function of the distance of the tDNA from the nuclear membrane.

My overall impression is that the work of Michieletto et al is a serious attempt to understand the basic principles behind how retroviral integration in a genome. The Authors are able to make interesting predictions based on their model, and are able to recapitulate existing experimental observation qualitatively and quantitatively. I believe this is quite remarkable given the simplicity of their model.

I have two concerns regarding this work. The first is that the title of the paper is somewhat misleading, because in the physical models that the Authors propose there is nothing specific about HIV.

The second concern I have is that the Authors have underestimated the importance of twisting and supercoiling in their physical modelling.

I believe this paper can be accepted in Nature Communications provided that the Authors can clarify these two important points, which I will illustrate below in detail.

I. SPECIFICITY OF HIV

The mechanisms described in the paper should be valid for any plasmid integration event in a genome. Therefore the biases described in this model should be valid for all retroviruses. I do not really know if this is true or there is strong evidence in favour or against this prediction.

I understand that the Authors significantly changed the manuscript to incorporate the suggestions in the previous review round to discuss the effect of the LEDGF/p75 factor. If I understand correctly, the Authors claim that there exists literature that proves that in the absence of such factor the integration bias towards euchromatin is still present. It is clear that in the absence of LEDGF/p75 the distribution of integration sites changes substantially. This to me then seems like the Authors are modeling the system in the absence of such factor. That is, the specificity of HIV integration patterns seem to be strongly influenced by the presence of LEDGF/p75, and when such specificity is lost, the model proposed in this work is valid. Following the same logic one would expect that also other retroviruses have exactly the same integration bias when their specific co-factors are knocked down. I am not aware that such studies exist, and it would be interesting to discuss the possibility of performing such experiments.

The main point I want to make here is that the title mentions that the study is about HIV, but I do not see why such model should be specific to HIV. A more relevant title would be “Physical Principles of Retroviral Integration in Eukaryotic Genomes”.

I see that this point was also raised by Reviewer #1 and that the Authors acknowledge this idea in the Discussion. The response by the Authors does not seem entirely satisfactory to me. This is because it give me the feeling that they consider their model as a first-order model for retroviral integration, and adding other details will result in a “refinement”, akin to a perturbation to the first-order model. I have not looked at the data from the papers cited in the manuscript, but from my experience I see that rarely in a biological system one can reason in these terms. All the factors and their interactions are strongly coupled together, and it is difficult to reason in terms of a main dominant effect (in this case, DNA elasticity) and a perturbation to that effect (specific co-factors). It seems to me that the Authors should stress this point, and highlight the important prediction that I mentioned above.

II. SUPERCOILING

The discussion about DNA supercoiling seems to me to be quite naive. Supercoiling is a very important effect and a highly nontrivial one. The linking number excess is accommodated by DNA in the form of bending and twisting, but the bending is **not** stored mainly in the apical loop as the Authors say. In fact, the famous bell-shaped curves in single-molecule DNA twisting experiments (height of the bead versus linking number excess) quite clearly demonstrate the opposite: it is the plectonemic part of the DNA molecule that accommodates almost all the excess linking number.

The Authors then discuss that their model predicts that integration probability should increase in supercoiled DNA, in line with experimental data. However, supercoiling effects are very far from trivial. An integration event would change the linking number locally, which would result in a complex scenario that would require a careful analysis. Is anything known about supercoiling of the HIV plasmid? If it is supercoiled (non-zero excess linking number), negatively and positively supercoiled host DNA would have opposite effects on integration probability.

Moreover, is it absolutely certain that HIV integrates as a plasmid? Can linear integration events occur?

Similarly to what discussed before, I do not believe that supercoiling can be considered like a future refinement of the model, but this time on purely physical terms. The twisting energy contributions

are of the same order of magnitude of the bending terms, and they change during the integration transactions, and therefore cannot be neglected.

I would suggest to rewrite the section "HIV Integration in Supercoiled DNA" or remove it entirely.

MINOR POINTS.

1. The fitting of the data of Marini et al. (figure 5F) is quite spectacular, but it seems that the model here contains many adjustable parameters. Can the Authors provide a measure of how good the fit is, taking into account the number of free parameters that they model?

2. In the modeling of euchromatin versus heterochromatin, the Authors choose to be euchromatic those regions of the genome that have a high GC-content and high transcriptional activity. They then demonstrate by simulations that the integration bias is uniform in a non-folded polymer and viceversa in a folded polymer. Is this prediction amenable to experimental verification? Do in vitro studies exist trying to prove or disprove the role of large-scale chromosome folding in HIV integration? It seems to me that the good correspondence that the Authors obtain with experimental data could be due to confounding factors that are related to GC-content, not to chromosome folding.

We thank the Reviewers for their attentive reading and the useful feedback. Below are our replies to their comments. Note that changes in the revision are highlighted in blue.

Reply to Reviewer #2:

##COMMENT

I think that the authors have replied in full details to the issues raised by referee #2 and #3. They have taken into account, in particular, all the useful comments and requests by referee #3. I still think the paper should be published.

REPLY

We are very grateful to the referee for her/his careful reading of our work and rebuttal and for recommending publication of our manuscript.

Reply to Reviewer #4:

COMMENT

The manuscript "Physical Principles of HIV Integration in the Human Genome" by Michieletto et al illustrates a study based on theoretical physics of how HIV integrates in the human genome. There are three main parts of the paper: in the first, the authors propose a quasi-equilibrium stochastic model based on computer simulations for HIV integration, and analyse the properties of physical integration of a plasmid in a model of a chromatin fibre; in the second, they go up a length scale and analyse within the same model how integration is biased towards different regions of the genome modelled as euchromatin or heterochromatin; finally, they propose a reaction-diffusion analytical framework to understand experimental patterns of HIV integration as a function of the distance of the genome from the nuclear membrane.

The main findings can be recapitulated as follows. First, the Authors are able to explain experimentally observed biases of HIV integration towards chromatin, flexible DNA regions, and curved DNA segments, because of straightforward energetic considerations. In their model in fact the elastic energy of the juxtaposed segments of the modelled polymers (vDNA for viral DNA, tDNA for target/host DNA) plays a pivotal role in determining the probability for an integration event. The second finding is that HIV prefers to integrate in euchromatic regions of the genome, because those regions are found in external "petal-like" structures which are readily accessible to the viral DNA. Finally, the Authors are able to reproduce accurately the patterns observed in a recent experimental work on the probability of HIV integration as a function of the distance of the tDNA from the nuclear membrane.

My overall impression is that the work of Michieletto et al is a serious attempt to understand the basic principles behind how retroviral integration in a genome. The Authors are able to make interesting predictions based on their model, and are able to recapitulate existing experimental observation qualitatively and quantitatively. I believe this is quite remarkable given the simplicity of their model.

REPLY

We thank the referee for his positive comments on our work.

COMMENT

I have two concerns regarding this work. The first is that the title of the paper is somewhat misleading, because in the physical models that the Authors propose there is nothing specific about HIV.

The second concern I have is that the Authors have underestimated the importance of twisting and supercoiling in their physical modelling.

I believe this paper can be accepted in Nature Communications provided that the Authors can clarify these two important points, which I will illustrate below in detail.

REPLY

We thank the referee for his consideration and for recommending publication subject to the clarification of these two points. We now constructively address his comments below and thus hope the amended version can now be published in Nature Communications.

COMMENT

I. SPECIFICITY OF HIV

The mechanisms described in the paper should be valid for any plasmid integration event in a genome. Therefore the biases described in this model should be valid for all retroviruses. I do not really know if this is true or there is strong evidence in favour or against this prediction.

REPLY

In the discussion section of the paper we highlight two classes of retroviruses (alpha and beta) which display peculiar integration patterns which markedly deviate from those of lentiviruses (such as HIV). In particular, we discuss that those two classes require the disruption of the nuclear membrane in order to infect the genome of the host and thus need to work during mitosis. For this reason, we believe that our current model (which focuses on interphase-like large-scale arrangements) cannot capture the patterns displayed by alpha and beta retroviruses. For all other classes, we agree with the referee that our model should be universal and indeed experimental evidence suggests that all these other classes display a (more or less pronounced) tendency to integrate in active regions of the genome (see review by Kvaratskhelia et al, Nucleic Acids Research, 2014).

Yet we should mention that the predictions of our models at both the mono-nucleosome (Fig.1) and full nuclear (Fig.5) scale could only be tested against HIV as this was the experimental system adopted for such experiments.

COMMENT

I understand that the Authors significantly changed the manuscript to incorporate the suggestions in the previous review round to discuss the effect of the LEDGF/p75 factor. If I understand correctly, the Authors claim that there exists literature that proves that in the absence of such factor the integration bias towards euchromatin is still present. It is clear that in the absence of LEDGF/p75 the distribution of integration sites changes substantially. This to me then seems like the Authors are modeling the system in the absence of such

factor. That is, the specificity of HIV integration patterns seem to be strongly influenced by the presence of LEDGF/p75, and when such specificity is lost, the model proposed in this work is valid. Following the same logic one would expect that also other retroviruses have exactly the same integration bias when their specific co-factors are knocked down. I am not aware that such studies exist, and it would be interesting to discuss the possibility of performing such experiments.

REPLY

As previously discussed, we acknowledge that LEDGF/p75 is an important factor that our model does not explicitly take into account. As highlighted in the Discussion, one possible and straightforward improvement in this direction is to consider specific binding sites for LEDGF/p75 along our model chromosomes. This detailed (and HIV-specific) study eludes the scope of the current work but it is mentioned in the “discussion” section of the current manuscript. In absence of these specific binding sites, we agree with the referee that our model should be understood as one capturing the behaviour of HIV integration in LEDGF/p75 knock-down cells (or in general retroviruses without their associated protein chaperones). We have now revised the MS in order to stress even more the general applicability of our model and the option of pursuing (simulated) knock-down experiments with other retroviruses. We thank the referee for the comment.

COMMENT

The main point I want to make here is that the title mentions that the study is about HIV, but I do not see why such model should be specific to HIV. A more relevant title would be “Physical Principles of Retroviral Integration in Eukaryotic Genomes”.

REPLY

We thank the referee for the suggestion which we have now taken on board.

COMMENT

I see that this point was also raised by Reviewer #1 and that the Authors acknowledge this idea in the Discussion. The response by the Authors does not seem entirely satisfactory to me. This is because it gives me the feeling that they consider their model as a first-order model for retroviral integration, and adding other details will result in a “refinement”, akin to a perturbation to the first-order model. I have not looked at the data from the papers cited in the manuscript, but from my experience I see that rarely in a biological system one can reason in these terms. All the factors and their interactions are strongly coupled together, and it is difficult to reason in terms of a main dominant effect (in this case, DNA elasticity) and a perturbation to that effect (specific co-factors). It seems to me that the Authors should stress this point, and highlight the important prediction that I mentioned above.

REPLY

To address the referee's concern, we stress here that there is strong evidence in the literature showing that knocking down specific co-factors still does **not** bring the integration of HIV to align with a random distribution. In particular, in Marshall et al, PloS One, 2007 the authors report that in LEDGF knock-down (KD) cells the "integration did not become random" and that "integration in transcription units [...] was still favoured, though to a reduced degree".

In a follow up study, Schrijvers et al, *Retrovirology*, 2012, the same group ascribed residual euchromatin affinity of HIV integration in LEDGF KD cells to another co-factor, HRP2. Yet, by scrutinising the data, one realises that even in double KD cells (LEDGF and HRP2) the distribution of integration events is still not statistically equivalent to that of a random process (see Fig. 3 of that paper). Therefore these data suggest that while LEDGF and other cofactors may be important, there must be some physical biases in the integration statistics, which it is important to characterise, but was not done previous to our work. The characterisation of these biases, which we argue purely originate from polymer physics, is the main aim of this work. We do not suggest that other effects are perturbatory or unimportant, and we have now made this clear in the Discussion.

COMMENT

II. SUPERCOILING

The discussion about DNA supercoiling seems to me to be quite naive. Supercoiling is a very important effect and a highly nontrivial one. The linking number excess is accommodated by DNA in the form of bending and twisting, but the bending is *not* stored mainly in the apical loop as the Authors say. In fact, the famous bell-shaped curves in single-molecule DNA twisting experiments (height of the bead versus linking number excess) quite clearly demonstrate the opposite: it is the plectonemic part of the DNA molecule that accommodates almost all the excess linking number.

REPLY

We certainly agree that supercoiling is an important and non-trivial effect, and indeed we have been working on several projects on this topic!

We feel that the point raised by the referee though originates from a misunderstanding. We agree that DNA supercoiling is partitioned in twist and writhe, however writhe (the non-local crossing of DNA's central axis) should not be taken as synonymous with bending. The referee is absolutely correct in saying that the linking number excess (supercoiling) is accommodated throughout the plectoneme, yet it can be shown that the variation of the tangent of the curve with respect to DNA's contour length ($|\frac{d\vec{t}}{ds}|$) is maximum at the apex of a plectoneme. To further support our argument, we point the referee to, e.g., papers of John Marko and co-workers (Dittmore et al, PRL 2017 or Brahmachari et al, PRE 2018) in which the authors show that plectonemes are nucleated and pinned at defects (mismatches or DNA weak spots) as these present a lower bending energy penalty. This argument is not dissimilar from the one we present in our manuscript, i.e. given that the apex of the plectoneme stores more bending energy than the rest of the plectoneme (which itself stores more bending energy than relaxed DNA), this is argued to be the most preferred location for integration events (in light of what discovered for mono-nucleosomes). We have now revised the relevant section of our MS in order to clarify the difference between bending and writhing.

COMMENT

The Authors then discuss that their model predicts that integration probability should increase in supercoiled DNA, in line with experimental data. However, supercoiling effects are very far from trivial. An integration event would change the linking number locally, which would result in a complex scenario that would require a careful analysis. Is anything known

about supercoiling of the HIV plasmid? If it is supercoiled (non-zero excess linking number), negatively and positively supercoiled host DNA would have opposite effects on integration probability.

REPLY

There is experimental evidence suggesting that when HIV enters the nucleus, it is chromatinised (see Knipe et al, *Virology*, 2013), and thus one may argue that it loses supercoiling (in the same way as eukaryotic chromatin is thought to harbour little levels of supercoiling). Yet, as far as we know, experiments in vitro may have the freedom to adjust the level of supercoiling of the viral plasmid and could thus address the role played by relative supercoiling in the integration process. We thank the referee for this interesting comment which we now include in the revised text.

COMMENT

Moreover, is it absolutely certain that HIV integrates as a plasmid? Can linear integration events occur?

REPLY

Topologically, a linear polymer recombining with another linear polymer (chromosome) would generate two disjoint polymers. Thus, linear integration is topologically not possible (and biologically not viable).

COMMENT

Similarly to what discussed before, I do not believe that supercoiling can be considered like a future refinement of the model, but this time on purely physical terms. The twisting energy contributions are of the same order of magnitude of the bending terms, and they change during the integration transactions, and therefore cannot be neglected. I would suggest to rewrite the section "HIV Integration in Supercoiled DNA" or remove it entirely.

REPLY

To address the referee's comment we point him to recent studies on the biochemical structures of the intasome (e.g., Lesbats et al, *Chemical Reviews*, 2016). These works suggest that during the process of retroviral integration, the twist of host and viral DNA needs to have the correct phasing in order to successfully carry out the integration. In this respect, the twist contribution should be considered as a "hard" constraint on the process rather than an elastic one. Overall, we agree with the referee that twist contributions are important (and worth considering in future improvements of the model), but we highlight that they may play a qualitatively different role with respect to bending contributions.

Nonetheless, several works on nicked DNA have confirmed that bending plays a major (and perhaps dominant) role in determining the bias in integration patterns (see e.g. Müller and Varmus, *EMBO J.*, 1994).

We finally thank the referee for his comments on this section. We have now taken on board several of his comments and revised the relevant part of the MS, in particular to make clear the distinction between bending and writhing and to more accurately emphasise the non-trivial nature of supercoiling which will require further work.

COMMENT

MINOR POINTS.

1. The fitting of the data of Marini et al. (figure 5F) is quite spectacular, but it seems that the model here contains many adjustable parameters. Can the Authors provide a measure of how good the fit is, taking into account the number of free parameters that they model?

REPLY

The model with fixed boundaries between regions has 2 free parameters: relative diffusion (D) and relative integration rates (K) of HIV in euchromatin-rich versus heterochromatin-rich zones. The choice of the heterochromatic diffusion coefficient $D_0=0.05 \text{ um}^2/\text{s}$ is based on experiments (Bosse et al, PNAS 2015) while the absolute integration rate $K_0=0.01 \text{ 1/s}$ is set so that the “penetration length” predicted by the analytical model, i.e. $\sqrt{D_0/K_0}$ is about 2.23 um and in qualitative agreement with the data from Marini et al. Different choices of this parameter would substantially change the simulated curve and bring it away from the broad experimental trend. For this reason, we choose to keep D_0 and K_0 fixed and vary K and D only.

To more quantitatively address the referee's comment we now provide in the revised SI a figure that shows the goodness of the fit (as chi-squared) in the parameter space (D,K). As discussed in the text, the fit seems robust and, pleasingly, finds its optimum in a region where the integration rate in euchromatin-rich zones is markedly larger than the one for heterochromatin (as expected from experimental consideration).

In the original MS, we then improved the model and considered “random” boundaries. Here, the amplitude of the randomisation (which sets the maximum displacement from the mean value) is also a free parameter which we now characterise more systematically in the revised SI. In particular, we show that the smoothing of the curves due to increasing A, can become detrimental (in terms of goodness of the fit) when too large. We thus also find an optimum in this parameter space. We thank the referee for suggesting this more thorough analysis of our model which will be useful for future studies.

COMMENT

2. In the modeling of euchromatin versus heterochromatin, the Authors choose to be euchromatic those regions of the genome that have a high GC-content and high transcriptional activity. They then demonstrate by simulations that the integration bias is uniform in a non-folded polymer and viceversa in a folded polymer. Is this prediction amenable to experimental verification? Do in vitro studies exist trying to prove or disprove the role of large-scale chromosome folding in HIV integration? It seems to me that the good correspondence that the Authors obtain with experimental data could be due to confounding factors that are related to GC-content, not to chromosome folding.

REPLY

We would gladly welcome in vitro experiments on the effect of large-scale chromosome folding on HIV integration. Yet we are not aware of any. In this respect, our simulations offer a testable prediction, i.e. that chromosome folding should play a major role in this context. We hope that this prediction will be directly tested in the near future by experimental teams working on the subject.

Reviewer #4 (Remarks to the Author):

The Authors have satisfactorily addressed my comments, and I therefore recommend publication of the manuscript in Nature Communications.

I did not understand only one of the replies, namely when they write "Topologically, a linear polymer recombining with another linear polymer (chromosome) would generate two disjoint polymers. Thus, linear integration is topologically not possible (and biologically not viable)."

If "A" is the vDNA, and "B" is the chromosome, why would it be topologically not possible to create a "B-A-B" configuration? Could the Authors clarify?

REVIEWERS' COMMENTS:

Reviewer #4 (Remarks to the Author):

The Authors have satisfactorily addressed my comments, and I therefore recommend publication of the manuscript in Nature Communications.

I did not understand only one of the replies, namely when they write "Topologically, a linear polymer recombining with another linear polymer (chromosome) would generate two disjoint polymers. Thus, linear integration is topologically not possible (and biologically not viable)."
If "A" is the vDNA, and "B" is the chromosome, why would it be topologically not possible to create a "B-A-B" configuration? Could the Authors clarify?

We thank the reviewer for the positive review. Regarding the final comment:

A reconnection between two linear polymers is shown in Fig1A of the SI. This figure should show, perhaps better than words, that two linear strands undergoing reconnections must generate two disjoint linear strands.

To put it another way, reconnections cannot alter the total number of bonds in the system. Thus, two reconnecting linear chains with N and M bonds (lengths $N+1$ and $M+1$) must generate two polymers with possibly different lengths but total bonds $N+M$. Thus, both must be linear and disjoint. Instead, the scenario envisaged by the referee brings in the reconnection event $N+M$ bonds and generates a system with $N+M+1$ bonds.